# A Compositional Atlas of Tractable Circuit Operations for Probabilistic Inference

**Antonio Vergari**[*]
School of Informatics
University of Edinburgh
avergari@ed.ac.uk

**YooJung Choi**
CS Department
UCLA
yjchoi@cs.ucla.edu

**Anji Liu**
CS Department
UCLA
liuanji@cs.ucla.edu

**Stefano Teso**
CS Department
University of Trento, Italy
stefano.teso@unitn.it

**Guy Van den Broeck**
CS Department
UCLA
guyvdb@cs.ucla.edu

## Abstract

Circuit representations are becoming the lingua franca to express and reason about tractable generative and discriminative models. In this paper, we show how complex inference scenarios for these models that commonly arise in machine learning—from computing the expectations of decision tree ensembles to information-theoretic divergences of sum-product networks—can be represented in terms of tractable modular operations over circuits. Specifically, we characterize the tractability of simple transformations—sums, products, quotients, powers, logarithms, and exponentials—in terms of sufficient structural constraints of the circuits they operate on, and present novel hardness results for the cases in which these properties are not satisfied. Building on these operations, we derive a unified framework for reasoning about tractable models that generalizes several results in the literature and opens up novel tractable inference scenarios.

## 1 Introduction

Many core computational tasks in machine learning (ML) and AI involve solving *complex integrals*, such as expectations appearing in training losses or in information-theoretic quantities including entropies or divergences. A fundamental question naturally arises: *under which conditions do these quantities admit tractable computation?* Or equivalently, when can we compute them *reliably and efficiently* without resorting to approximations or heuristics? If we are able to find model classes to tractably compute these quantities of interest—henceforth called *queries*—we can then design efficient algorithms with important applications in learning, approximate inference [44], model compression [28], explainable AI [24, 47, 52] and algorithmic bias detection [23, 7, 6].

This "quest" for tracing the tractability of different queries has been carried out several times, often independently for different model classes in ML and AI and crucially, for each query in isolation. For example, the computation of the Kullback-Leibler divergence (KLD) is known to have a closed form for Gaussians, but only recently has an exact algorithm been derived for a more complex tractable model class such as probabilistic sentential decision diagrams (PSDDs) [28]. On the other hand, tractable computation of the entropy, despite being a sub-routine for the KLD, has only been derived for a different tractable model class—selective sum-product networks (SPNs) [36]—by Shih and Ermon [44]. In the current paradigm, if one were to trace the tractability of a query that has not

---

[*]This work was done while AV was at the CS Department at UCLA.

35th Conference on Neural Information Processing Systems (NeurIPS 2021).

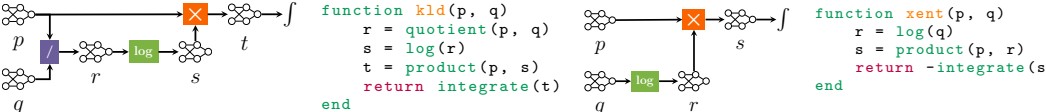

```
function kld(p, q)
    r = quotient(p, q)
    s = log(r)
    t = product(p, s)
    return integrate(t)
end
```

```
function xent(p, q)
    r = log(q)
    s = product(p, r)
    return -integrate(s)
end
```

Figure 1: Computational pipelines of the KLD (left) and cross entropy (right) over two distributions $p$ and $q$ encoded as circuits, with the intermediate computations ($r$, $s$ and $t$) also represented as circuits. Their corresponding implementations in a few lines of Julia code are shown on their right.

yet been investigated but still involves the same "building blocks" such as logarithms, integrals and products over distributions, for instance Rényi's alpha divergence [40], they would need to derive a novel custom algorithm for each model class and prove its tractability from scratch.

In this paper, we take a different path and introduce a general framework under which the tractability of complex queries can be traced in a *unified and effortless manner over model classes and query classes*. To abstract from the different model formalisms, we carry our analysis over circuit representations [8] as they subsume many tractable generative models—probabilistic circuits such as Chow-Liu trees [9], hidden Markov models (HMMs) [39], sum-product networks (SPNs) [38], and other deep mixture models—as well as discriminative ones, including decision trees [25, 10] and deep regressors [23], thus enabling a unified treatment across model classes.

To generalize our analysis across queries, we propose to represent a single query as a *circuit pipeline*: a computational graph whose intermediate operations transform and combine the input circuits into other circuits. We can first build a set of simple tractable circuit transformations—sums, products, powers, logarithms, and exponentials—and then i) analyze the tractability of a single query by propagating the sufficient conditions for tractability of the intermediate operators in the pipeline; and ii) automatically distill a tractable inference algorithm by composing the operators used. For instance, Fig. 1 shows the pipeline for computing the KLD of $p$ and $q$, two distributions encoded by circuits. We can identify a general class of models that supports its tractable computation: by tracing the conditions for tractable quotient, logarithm, and product over circuits such that the output circuit (i.e., $t$) admits tractable integration, we can derive a set of sufficient conditions for the input circuits. Moreover, we can *reuse* the logarithm and product operations in the KLD pipeline to reason about the tractability of cross entropy, in the very same way we can reuse the corresponding subroutines we provide in Julia to quickly implement algorithms for the two queries in a couple lines of code as shown in Fig. 1. This compositionality greatly speeds up the design of novel tractable algorithms.

We make the following contributions: (1) a systematic way to compositionally answer many complex queries using simple circuit transformations (Sec. 3), proving sufficient conditions for their tractability and computational hardness when these conditions are unmet (Tab. 1); (2) a unification and generalization of many inference algorithms proposed in the literature so far for specific representations (Sec. 4); (3) novel tractability and hardness results of complex information-theoretic queries including several widely used entropies and divergences (Tab. 2); and (4) a publicly available implementation of these operators in the Juice circuit library [12]. We now start by introducing the circuit language.

## 2  Circuit Representations

Circuits represent functions as parameterized computational graphs. By imposing certain structural constraints on these graphs, we can guarantee the tractability of certain operations over the encoded functions. Moreover, these constraints help understand how *circuits unify several classical tractable model classes*, such as mixture models, bounded-treewidth probabilistic graphical models (PGMs), decision trees, and compact logical function representations [8, 51]. As such, circuits provide *a language for building and reasoning about tractable representations*, and it follows that all our results in the following sections automatically translate to these model classes.

We introduce the basic rules of this language by distinguishing between general circuits and those encoding probability distributions, as some operators in Sec. 3 may be restricted to the latter. Then, we will review the structural constraints we need to characterize different inference scenarios, also known as classes of queries. We denote random variables by uppercase letters ($X$) and their assignments by lowercase ones ($x$). Sets of variables and their assignments are denoted by bold uppercase ($\mathbf{X}$) and bold lowercase ($\boldsymbol{x}$) letters, respectively, and the set of all their values as $\mathsf{val}(\mathbf{X})$.

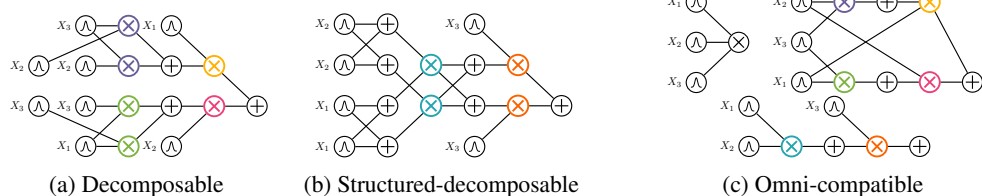

|  |  |  |
|---|---|---|
| (a) Decomposable | (b) Structured-decomposable | (c) Omni-compatible |

Figure 2: Examples of circuits with different structural properties. The feedforward order is from left to right; input units are labeled by their scopes; and sum parameters are omitted for visual clarity. Product units of the rearranged omni-compatible circuits encoding $p(X_1) \cdot p(X_2) \cdot p(X_3)$ are shown in (c) and color-coded with those of matching scope in (a) and (b).

**Definition 2.1** (Circuit). A circuit $p$ over variables $\mathbf{X}$ is a parameterized computational graph encoding a function $p(\mathbf{X})$ and comprising three kinds of computational units: *input*, *product*, and *sum*. Each inner unit $n$ (i.e., product or sum unit) receives inputs from other units, denoted $\mathsf{in}(n)$. If $n$ is an input unit, it encodes a parameterized function $p_n(\phi(n))$ over variables $\phi(n) \subseteq \mathbf{X}$, also called its *scope*. Instead, if $n$ is a sum unit, it encodes $\sum_{c \in \mathsf{in}(n)} \theta_c p_c(\phi(c))$ where $\theta_c \in \mathbb{R}$ are the sum parameters; while if it is a product unit, it encodes $\prod_{c \in \mathsf{in}(n)} p_c(\phi(c))$. The scope of an inner unit is the union of the scopes of its inputs: $\phi(n) = \bigcup_{c \in \mathsf{in}(n)} \phi(c)$. The output unit of the circuit is the last unit (i.e., with out-degree 0) in the graph, encoding $p(\mathbf{X})$. The *support* of $p$ is the set of all complete states for $\mathbf{X}$ for which the output of $p$ is non-zero: $\mathsf{supp}(p) = \{\boldsymbol{x} \in \mathsf{val}(\mathbf{X}) \mid p(\boldsymbol{x}) \neq 0\}$.

Circuits can be understood as compact representations of polynomials with exponentially many terms, whose indeterminates are the functions encoded by the input units. These functions are assumed to be simple enough to allow tractable computations of the operations discussed in this paper. Fig. 2 shows some examples of circuits. A *probabilistic circuit* (PC) [8] represents a (possibly unnormalized) probability distribution by encoding its probability mass, density, or a combination thereof.

**Definition 2.2** (Probabilistic circuit). A PC over variables $\mathbf{X}$ is a circuit encoding a function $p$ that is non-negative for all values of $\mathbf{X}$; i.e., $\forall \boldsymbol{x} \in \mathsf{val}(\mathbf{X}) : p(\boldsymbol{x}) \geq 0$.

From here on, we will assume that a PC has positive sum parameters and input units that model valid (unnormalized) distributions, which is a sufficient condition to satisfy the above definition. Moreover, w.l.o.g. we will assume that each layer of a circuit alternates between sum and product units and that every product unit $n$ receives only two inputs $c_1, c_2$, i.e., $p_n(\mathbf{X}) = p_{c_1}(\mathbf{X}) \cdot p_{c_2}(\mathbf{X})$. These conditions can easily be enforced on a circuit in exchange for only a polynomial increase in its size [49, 50].

Computing (functions of) $p(\mathbf{X})$, or in other words performing *inference*, can be done by evaluating its computational graph. Hence, the computational cost of inference on a circuit is a function of its *size*, defined as the number of edges in it and denoted as $|p|$. For instance, querying the value of $p$ for a complete assignment $\boldsymbol{x}$ equals its *feedforward* evaluation—inputs before outputs—and therefore is linear in $|p|$. Other common inference scenarios such as function integration—which translate to *marginal inference* in the context of probability distributions—can be tackled in linear time with circuits that exhibit certain structural properties, as discussed next.

**Structural Properties of Circuits.** Structural constraints on the computational graph of a circuit in terms of its scope or support provide sufficient and/or necessary conditions for certain queries to be tractably computed. We now define the structural properties needed for the query classes that this work will focus on, referring to Choi et al. [8] for more details.

**Definition 2.3** (Smoothness). A circuit is *smooth* if for every sum unit $n$, its inputs depend on the same variables: $\forall c_1, c_2 \in \mathsf{in}(n), \phi(c_1) = \phi(c_2)$.

Smooth PCs generalize shallow mixture models [30] to deep and hierarchical models. For instance, a Gaussian mixture model (GMM) can be represented as a smooth PC with a single sum unit over as many input units as mixture components, each encoding a (multivariate) Gaussian density.

**Definition 2.4** (Decomposability). A circuit is *decomposable* if the inputs of every product unit $n$ depend on disjoint sets of variables: $\mathsf{in}(n) = \{c_1, c_2\}, \phi(c_1) \cap \phi(c_2) = \emptyset$.

Decomposable product units encode local factorizations. That is, a decomposable product unit $n$ over variables $\mathbf{X}$ encodes $p_n(\mathbf{X}) = p_1(\mathbf{X}_1) \cdot p_2(\mathbf{X}_2)$ where $\mathbf{X}_1$ and $\mathbf{X}_2$ form a partition of $\mathbf{X}$. Taken together, decomposability and smoothness are a sufficient and necessary condition for performing tractable integration over arbitrary sets of variables in a single feedforward pass, as they enable larger integrals to be efficiently decomposed into smaller ones [15, 8]. Next proposition formalizes it.

**Proposition 2.1** (Tractable integration, Choi et al. [8])**.** *Let $p$ be a smooth and decomposable circuit over $\mathbf{X}$ with input functions that can be tractably integrated. Then for any variables $\mathbf{Y} \subseteq \mathbf{X}$ and their assignment $\boldsymbol{y}$, the integral $\int_{\boldsymbol{z} \in \mathsf{val}(\mathbf{Z})} p(\boldsymbol{y}, \boldsymbol{z}) d\mathbf{Z}$ can be computed exactly in $\Theta(|p|)$ time, where $\mathbf{Z}$ denotes $\mathbf{X} \setminus \mathbf{Y}$.*

As the complex queries we focus on in this work involve integration as the last step, it is therefore needed that any intermediate operation preserves at least decomposability; smoothness is less of an issue, as it can be enforced in polytime [45]. A key additional constraint over scope decompositions is *compatibility*. Intuitively, two decomposable circuits are compatible if they can be rearranged in polynomial time[2] such that their respective product units, once matched by scope, decompose in the same way. We formalize this with the following inductive definition.

**Definition 2.5** (Compatibility)**.** Two circuits $p$ and $q$ over variables $\mathbf{X}$ are *compatible* if (1) they are smooth and decomposable and (2) any pair of product units $n \in p$ and $m \in q$ with the same scope can be rearranged into binary products that are mutually compatible and decompose in the same way: $(\phi(n) = \phi(m)) \implies (\phi(n_i) = \phi(m_i),\ n_i$ and $m_i$ are compatible) for some rearrangement of the inputs of $n$ (resp. $m$) into $n_1, n_2$ (resp. $m_1, m_2$).

We can derive from compatibility the following properties pertaining to a single circuit, which will be useful in our analysis later.

**Definition 2.6** (Special types of compatibility)**.** A circuit is *structured-decomposable* if it is compatible with itself. A decomposable circuit $p$ over $\mathbf{X}$ is *omni-compatible* if it is compatible with any smooth and decomposable circuit over $\mathbf{X}$.

Not all decomposable circuits are structured-decomposable (see Figs. 2a and 2b), but some can be rearranged to be compatible with any decomposable circuit. For instance, in Fig. 2c, the fully factorized product unit $p(\mathbf{X}) = p_1(X_1) \cdot p_2(X_2) \cdot p_3(X_3)$ can be rearranged into $p_1(X_1) \cdot (p_2(X_2) \cdot p_3(X_3))$ and $p_2(X_2) \cdot (p_1(X_1) \cdot p_3(X_3))$ to match the yellow and pink products in Fig. 2a. We can easily see that omni-compatible circuits must assume the form of mixtures of fully-factorized models; i.e., $\sum_i \theta_i \prod_j p_{i,j}(X_j)$. For example, an additive ensemble of decision trees over variables $\mathbf{X}$ can be represented as an omni-compatible circuit (see Ex. D.1 in the Appendix). Also note that if two circuits are compatible and neither is omni-compatible, then both must be structured decomposable.

**Definition 2.7** (Determinism)**.** A circuit is *deterministic* if the inputs of every sum unit $n$ have disjoint supports: $\forall c_1, c_2 \in \mathsf{in}(n), c_1 \neq c_2 \implies \mathsf{supp}(c_1) \cap \mathsf{supp}(c_2) = \emptyset$.

Analogously to decomposability, determinism induces a recursive partitioning, but this time over the support of a circuit. For a deterministic sum unit $n$, the partitioning of its support can be made explicit by introducing an indicator function per each of its inputs, i.e., $\sum_{c \in \mathsf{in}(n)} \theta_c p_c(\boldsymbol{x}) = \sum_{c \in \mathsf{in}(n)} \theta_c p_c(\boldsymbol{x}) [\![\boldsymbol{x} \in \mathsf{supp}(p_c)]\!]$. Determinism allows for tractable maximization of circuits [13, 8]. While we do not consider maximization queries in this work, determinism will still play a crucial role in the next sections. Moreover, bounded-treewidth PGMs, such as Chow-Liu trees [9] and thin junction trees [1], can efficiently be represented as smooth, deterministic, and decomposable PCs via *compilation* [13, 11]. Probabilistic sentential decision diagrams (PSDDs) [26] are deterministic and structured-decomposable PCs that can be efficiently learned from data [11].

# 3 From Simple Circuit Transformations...

This section aims to build an atlas of simple operations over circuits which can then be composed into more complex queries via circuit pipelines—computational graphs whose units are tractable operators over circuits. To compose two operators, we would need that the output circuits of one satisfy the structural properties required for the inputs of the other. As such, for each of these operations we are interested in characterizing (1) its tractability in terms of the structural properties of its input circuits,

---

[2]By changing the order in which n-ary product units are turned into a series of binary product units.

Table 1: ***Tractability and hardness of simple circuit operations***. Tractable properties on inputs translate to properties on outputs. E.g., for the quotient $p/q$, if $p$ and $q$ are compatible (Cmp) and $q$ is deterministic (Det), then the output is decomposable (Dec); also (+) deterministic if $p$ is deterministic; and structured-decomposable (SD) if both $p$ and $q$ are. Hardness results are for representing the output as a smooth (Sm) and decomposable circuit without some input condition.

| Operation | | Tractability | | | Hardness | |
|---|---|---|---|---|---|---|
| | | Input properties | Output properties | Time Complexity | | |
| SUM | $\theta_1 p + \theta_2 q$ | (+Cmp) | (+SD) | $\mathcal{O}(|p|+|q|)$ (Prop. B.1) | NP-hard for Det output [43] | |
| PRODUCT | $p \cdot q$ | Cmp (+Det, +SD) | Dec (+Det, +SD) | $\mathcal{O}(|p||q|)$ (Thm. 3.2) | #P-hard w/o Cmp | (Thm. 3.1) |
| POWER | $p^n, n \in \mathbb{N}$ | SD (+Det) | SD (+Det) | $\mathcal{O}(|p|^n)$ (Thm. 3.3) | #P-hard w/o SD | (Thm. 3.3 and B.1) |
| | $p^\alpha, \alpha \in \mathbb{R}$ | Sm, Dec, Det (+SD) | Sm, Dec, Det (+SD) | $\mathcal{O}(|p|)$ (Thm. 3.5) | #P-hard w/o Det | (Thm. 3.4) |
| QUOTIENT | $p/q$ | Cmp; $q$ Det (+$p$ Det,+SD) | Dec (+Det,+SD) | $\mathcal{O}(|p||q|)$ (Thm. B.3) | #P-hard w/o Det | (Thm. B.2) |
| LOG | $\log(p)$ | Sm, Dec, Det | Sm, Dec | $\mathcal{O}(|p|)$ (Thm. 3.6) | #P-hard w/o Det | (Thm. 3.6) |
| EXP | $\exp(p)$ | linear | SD | $\mathcal{O}(|p|)$ (Prop. 3.1) | #P-hard | (Thm. 3.7) |

and (2) its closure w.r.t. these properties, i.e. whether they are preserved in the output circuit, in order to compose many operations together in a pipeline, while (3) providing an efficient algorithmic implementation for it. As we are interested in pipelines for queries involving integration, we would expect the output circuits to at least retain decomposability (see Prop. 2.1). For a pipeline in which all operators can be computed tractably, a simple tractable algorithm can be then distilled for it. Furthermore, our analysis will highlight if one needs to resort to approximations, by tracing the hardness of representing the output of an operator as a decomposable circuit when some property of its inputs is unmet. Tab. 1 summarizes all our results. For space constraints, we discuss the main theorems next and report their complete statements and proofs in the Appendix.

**Sum of Circuits.** The operation of summing two circuits $p(\mathbf{Z})$ and $q(\mathbf{Y})$ is defined as $s(\mathbf{X}) = \theta_1 \cdot p(\mathbf{Z}) + \theta_2 \cdot q(\mathbf{Y})$ for $\mathbf{X} = \mathbf{Z} \cup \mathbf{Y}$ and two real parameters $\theta_1, \theta_2 \in \mathbb{R}$. This operation, which is at the core of additive ensembles of tractable representations,[3] can be realized by introducing a single sum unit that takes as input $p$ and $q$. Summation applies to any input circuits, regardless of structural assumptions, and it preserves several properties (see Prop. B.1 in the Appendix). In particular, if $p$ and $q$ are decomposable then $s$ is also decomposable; moreover, if they are compatible then $s$ is structured-decomposable as well as compatible with $p$ and $q$. However, representing a sum as a deterministic circuit is known to be NP-hard [43], even for compatible and deterministic inputs.

**Product of Circuits.** The product of two circuits $p(\mathbf{Z})$ and $q(\mathbf{Y})$ can be expressed as $m(\mathbf{X}) = p(\mathbf{Z}) \cdot q(\mathbf{Y})$ for variables $\mathbf{X} = \mathbf{Z} \cup \mathbf{Y}$. If $\mathbf{Z}$ and $\mathbf{Y}$ are disjoint, the product $m$ is already decomposable. Otherwise, Shen et al. [43] proved that representing the product of two decomposable circuits as a decomposable circuit is NP-hard, even if they are deterministic. We prove a novel result: it is even #P-hard to multiply two structured-decomposable and deterministic circuits.

**Theorem 3.1** (Hardness of product). *If $p$ and $q$ are two structured-decomposable and deterministic circuits, then computing their product as a decomposable circuit is #P-hard.*

Shen et al. [43] also introduced an efficient algorithm for the product of two structured-decomposable and deterministic PCs that are compatible (namely PSDDs). We generalize this result by proving that compatibility alone is sufficient for the tractable product computation of any two circuits.

**Theorem 3.2** (Tractable product). *If $p$ and $q$ are two compatible circuits, then computing their product as a decomposable circuit that is compatible with them can be done in $\mathcal{O}(|p|\,|q|)$ time.*

The proof is by construction and leads to Alg. 3 in the Appendix. In the following, we provide a sketch of the algorithm for the case $\mathbf{X} = \mathbf{Z} = \mathbf{Y}$. Intuitively, the idea is to "break down" the construction of the product circuit in a recursive manner by exploiting compatibility. The base case is where $p$ and $q$ are input units with simple parametric forms. Their product can be represented as a single input unit as long as we can find a simple parametric form for it, as is the case for products of exponential families such as (multivariate) Gaussians. Next, we consider the inductive steps where $p$ and $q$ are two sum or product units. If $p$ and $q$ are compatible product units, they decompose $\mathbf{X}$ the same way for some ordering of inputs; i.e., $p(\mathbf{X}) = p_1(\mathbf{X}_1)p_2(\mathbf{X}_2)$ and $q(\mathbf{X}) = q_1(\mathbf{X}_1)q_2(\mathbf{X}_2)$. Then, their product $m$ as a decomposable circuit can be constructed recursively from the products of their inputs: $m(\mathbf{X}) = (p_1 q_1)(\mathbf{X}_1) \cdot (p_2 q_2)(\mathbf{X}_2)$. On the other hand, if $p$ and $q$ are smooth sum units, written as $p(\mathbf{X}) = \sum_i \theta_i p_i(\mathbf{X})$ and $q(\mathbf{X}) = \sum_j \theta'_j q_j(\mathbf{X})$, we can obtain their product $m$ recursively

---
[3]If $p$ and $q$ are PCs, then $s$ is a PC encoding a monotonic mixture model if $\theta_1, \theta_2 > 0$ and $\theta_1 + \theta_2 = 1$.

by distributing product over sum. In other words, $m(\mathbf{X}) = \sum_{i,j} \theta_i \theta_j' (p_i q_j)(\mathbf{X})$. Note that if both input circuits are also deterministic, $m$ is also deterministic since $\mathsf{supp}(p_i q_j) = \mathsf{supp}(p_i) \cap \mathsf{supp}(q_j)$ are disjoint for different $i, j$. Combining these, the algorithm will recursively compute the product of each pair of units in $p$ and $q$ with matching scopes. Assuming efficient products for input units, the overall complexity is $\mathcal{O}(|p|\,|q|)$, which yields a compact circuit $m$ of size $\mathcal{O}(|p|\,|q|)$. This upper bound is loose and in practice product circuits will be much smaller as our experiments show (Sec. 5), especially if inputs are deterministic as products of units with disjoint supports will be "pruned" away.

**Powers of a Circuit.**    The $\alpha$-power of a PC $p(\mathbf{X})$ for an $\alpha \in \mathbb{R}$ is denoted as $p^\alpha(\mathbf{X})$ and is an operation needed to compute generalizations of the entropy of a PC and related divergences (Sec. 4). Let us first consider natural powers ($\alpha \in \mathbb{N}$) which can be computed even for general circuits.

**Theorem 3.3** (Natural powers). *If $p$ is a structured-decomposable circuit, then for any $\alpha \in \mathbb{N}$, its power can be represented as a structured-decomposable circuit in $\mathcal{O}(|p|^\alpha)$ time. Otherwise, if $p$ is only smooth and decomposable, then computing $p^\alpha(\mathbf{X})$ as a decomposable circuit is #P-hard.*

The proof for tractability easily follows by directly applying the product operation repeatedly. However, the exponential dependence on $\alpha$ is unavoidable unless P=NP as we demonstrate in Thm. B.1 in the Appendix, thus rendering the operation intractable for large $\alpha$.

Turning our attention to non-natural $\alpha \in \mathbb{R}$, and restricting our attention to PCs, structured-decomposability is not sufficient to tractably compute $\alpha$-powers, which we will show in the next theorem for $\alpha = -1$. First, as zero raised to a negative power is undefined, we instead consider the *restricted $\alpha$-power* of a PC, denoted as $p^\alpha(\boldsymbol{x})\big|_{\mathsf{supp}(p)}$ and equal to $(p(\boldsymbol{x}))^\alpha$ if $\boldsymbol{x} \in \mathsf{supp}(p)$ and 0 otherwise. Note that this is equivalent to the $\alpha$-power if $\alpha \geq 0$. Abusing notation, we will also denote this by $p^\alpha(\boldsymbol{x})[\![\boldsymbol{x} \in \mathsf{supp}(p)]\!]$, where $[\![\cdot]\!]$ stands for indicator functions.

**Theorem 3.4** (Hardness of reciprocals). *If $p$ is a structured-decomposable circuit over variables $\mathbf{X}$, then computing $p^{-1}(\mathbf{X})\big|_{\mathsf{supp}(p)}$ as a decomposable circuit is #P-hard.*

The key property that enables efficient computation of power circuits is determinism. More interestingly, we do not require structured-decomposability, but only smoothness and decomposability.

**Theorem 3.5** (Tractable real powers). *If $p$ is a smooth, decomposable, and deterministic PC, then for any $\alpha \in \mathbb{R}$, its restricted power can be represented as a smooth, decomposable, and deterministic circuit that is compatible with $p$ in $\mathcal{O}(|p|)$ time.*

Again, the proof is done by construction and detailed in Sec. B.3.   The key insight is that restricted powers "break down" over a smooth and deterministic sum unit $p$.   That is, $\left(\sum_i \theta_i p_i(\boldsymbol{x})[\![\boldsymbol{x} \in \mathsf{supp}(p_i)]\!]\right)^\alpha [\![\boldsymbol{x} \in \mathsf{supp}(p)]\!] = \sum_i \theta_i^\alpha p_i^\alpha(\boldsymbol{x})[\![\boldsymbol{x} \in \mathsf{supp}(p_i)]\!]$. This follows from the fact that for any $\boldsymbol{x}$, at most one indicator $[\![\boldsymbol{x} \in \mathsf{supp}(p_i)]\!]$ evaluates to 1. As such, when multiplying a deterministic sum unit with itself, each input will only have overlapping support with itself, thus effectively matching product units only with themselves. This is why decomposability suffices. In conclusion, this recursive decomposition of the power of a circuit will result in the power circuit having the same structure as the original circuit, with input functions and sum parameters replaced by their $\alpha$-powers. The space and time complexity of the algorithm is $\mathcal{O}(|p|)$ for smooth, deterministic, and decomposable PCs, even for natural powers. This will be a key insight to compactly multiply circuits with the same support structure, such as when computing logarithms and entropies (Sec. 4).

We can already see an example of how simple operators can be composed to derive other tractable ones. Consider the quotient of two circuits $p(\mathbf{X})$ and $q(\mathbf{X})$, denoted as $p(\mathbf{X})/q(\mathbf{X})$, and restricted to $\mathsf{supp}(q)$. The quotient, appearing in queries such as KLD or Itakura-Saito divergence (Sec. 4), can be computed by first taking the reciprocal circuit (i.e., the $(-1)$-power) of $q$, followed by its product with $p$. Thus, if $q$ is deterministic and compatible with $p$, we can take its reciprocal—which will have the same structure as $q$—and multiply with $p$ to obtain the quotient as a decomposable circuit as we show in Thm. B.3 in the Appendix. There, Thm. B.2 instead proves that the quotient between $p$ and a non-deterministic $q$ is #P-hard even if they are compatible.

**Logarithms of a PC.** The logarithm of a PC $p(\mathbf{X})$, denoted $\log p(\mathbf{X})$, is fundamental in computing quantities such as entropies and divergences between distributions (Sec. 4). Since the log is undefined for 0 we will again consider the *restricted logarithm*, denoted as $\log p(\boldsymbol{x})\big|_{\mathsf{supp}(p)}$ and equal to $\log p(\boldsymbol{x})$ if $\boldsymbol{x} \in \mathsf{supp}(p)$ and 0 otherwise.

**Theorem 3.6** (Logarithms)**.** *If $p$ is a smooth, deterministic and decomposable PC, then its restricted logarithm $\log p(\mathbf{X})|_{\mathsf{supp}(p)}$ can be represented as a decomposable circuit in $\mathcal{O}(|p|)$ time. Otherwise, if $p$ is only smooth and decomposable, or even structured-decomposable, computing its restricted logarithm as a decomposable circuit is #P-hard.*

Note that while the input of the logarithm operator must be a PC, its output can be a general circuit. Moreover, if $p$ is structured decomposable, then so is its logarithm. The tractability proof is again by construction and is detailed in Sec. B.5 in the Appendix. We point out that determinism again allows the restricted log to decompose over the support of the PC, but this time the output circuit is *not* deterministic, as more than one of its inputs can yield a non-zero output at a time. Nevertheless, the inputs of the newly introduced sum units can be clearly partitioned into groups sharing the same support of the corresponding product units in $p$. This acts as a relaxed form of determinism when at most three inputs can be non-zero at once, and it implies that whenever we multiply a deterministic circuit and its logarithmic circuit—for instance to compute its Shannon entropy (Sec. 4)—we can leverage the sparsifying effect of non-overlapping supports and perform only a linear number of products (cf. product and power operators). We confirm this empirically in Sec. 5 when evaluating the size of the intermediate circuits for computing entropy pipelines over real-world distributions.

**Exponentials of a Circuit.**   The exponential of a circuit $p(\mathbf{X})$, denoted $\exp(p(\mathbf{X}))$, is the inverse operation of the logarithm and is a fundamental operation when representing distributions such as log-linear models [27]. Similarly to the logarithm, building a decomposable circuit that encodes an exponential of a circuit is hard in general.

**Theorem 3.7** (Hardness of exponentials)**.** *If $p$ is a smooth and decomposable circuit, then, computing its exponential as a decomposable circuit is #P-hard, even if $p$ is structured-decomposable.*

Unlike the logarithm however, restricting the operation to deterministic circuits does not help with tractability, since the issue comes from product units: the exponential of a product is neither a sum nor product of exponentials. Nevertheless, it is easy to see that if $p$ encodes a linear sum over its variables, i.e., $p(\mathbf{X}) = \sum_i \theta_i X_i$, we could easily represent its exponential as a circuit comprising a single decomposable product unit, hence tractably.

**Proposition 3.1** (Tractable exponential of a linear circuit)**.** *If $p$ is a linear circuit, then its exponential can be represented as an omni-compatible circuit in $\mathcal{O}(|p|)$ time.*

Note that if we were to add an additional deterministic sum unit over many omni-compatible circuits built in this way, we would retrieve a mixture of truncated exponentials [32, 55]. This is the largest class of tractable exponentials we know so far, and enlarging its boundaries is an open problem.

**More operators?**   Our compositional atlas is now complete. In fact, if we were to add an additional circuit operator to the atlas, it would have to take the form of the already discussed operators. First, we require from any functional $f$ to be applied to a circuit $p(\mathbf{X})$ to yield a smooth and decomposable circuit $f(p(\mathbf{X}))$ in order to admit tractable integration and to be added to our atlas. To that end, as usual we can assume to apply $f$ to the input units of $p$ and obtain tractable representations for the new input units; this is generally the case for simple parametric input functions. Next, we would require $f$ to decompose over products and over sums. In other words, we first need that $f(p_1(\mathbf{X}_1) \cdot p_2(\mathbf{X}_2))$ can be broken down to either a product $f(p_1(\mathbf{X}_1)) \cdot f(p_2(\mathbf{X}_2))$ or sum $f(p_1(\mathbf{X}_1)) + f(p_2(\mathbf{X}_2))$. Furthermore, we want $f$ to similarly decompose over sum units; that is, $f(p_1(\mathbf{X}_1) + p_2(\mathbf{X}_2))$ also yields a product or sum of $f(p_1(\mathbf{X}_1))$ and $f(p_2(\mathbf{X}_2))$. As the next lemma states, a non-linear function $f$ that satisfies either of the above two conditions must be a power, logarithmic, or exponential function.

**Lemma 3.8.** *Let $f$ be a continuous function. If $f : \mathbb{R} \to \mathbb{R}$ satisfies $f(x + y) = f(x) + f(y)$ then it is a linear function $\beta \cdot x$; if $f : \mathbb{R}_+ \to \mathbb{R}_+$ satisfies $f(x \cdot y) = f(x) \cdot f(y)$, then it takes the form $x^\beta$; if instead $f : \mathbb{R}_+ \to \mathbb{R}$ satisfies $f(x \cdot y) = f(x) + f(y)$, then it takes the form $\beta \log(x)$; and if $f : \mathbb{R} \to \mathbb{R}_+$ satisfies that $f(x + y) = f(x) \cdot f(y)$ then it is of the form $\exp(\beta \cdot x)$, for a certain $\beta \in \mathbb{R}$.*

## 4   … to Complex Compositional Queries

In this section, we show how our atlas of simple tractable operators can be effectively used to systematically find a tractable model class for *any advanced query that comprises these operators.*

Table 2: *Tractability and hardness of information-theoretic queries over circuits.* Tractability given some conditions over the input circuits; computational hardness when some of these are unmet.

| | Query | Tract. Conditions | Hardness | Reference |
|---|---|---|---|---|
| CROSS ENTROPY | $-\int p(\boldsymbol{x})\log q(\boldsymbol{x})\,d\mathbf{X}$ | Cmp, q Det | #P-hard w/o Det | Thm. C.1 |
| SHANNON ENTROPY | $-\sum p(\boldsymbol{x})\log p(\boldsymbol{x})$ | Sm, Dec, Det | coNP-hard w/o Det | Thm. C.2 |
| RÉNYI ENTROPY | $(1-\alpha)^{-1}\log\int p^\alpha(\boldsymbol{x})\,d\mathbf{X}, \alpha\in\mathbb{N}$ | SD | #P-hard w/o SD | Thm. C.5 |
| | $(1-\alpha)^{-1}\log\int p^\alpha(\boldsymbol{x})\,d\mathbf{X}, \alpha\in\mathbb{R}_+$ | Sm, Dec, Det | #P-hard w/o Det | Thm. C.6 |
| MUTUAL INFORMATION | $\int p(\boldsymbol{x},\boldsymbol{y})\log(p(\boldsymbol{x},\boldsymbol{y})/(p(\boldsymbol{x})p(\boldsymbol{y})))$ | Sm, SD, Det* | coNP-hard w/o SD | Thm. C.3 |
| KULLBACK-LEIBLER DIV. | $\int p(\boldsymbol{x})\log(p(\boldsymbol{x})/q(\boldsymbol{x}))d\mathbf{X}$ | Cmp, Det | #P-hard w/o Det | Thm. C.4 |
| RÉNYI'S ALPHA DIV. | $(1-\alpha)^{-1}\log\int p^\alpha(\boldsymbol{x})q^{1-\alpha}(\boldsymbol{x})\,d\mathbf{X}, \alpha\in\mathbb{N}$ | Cmp, q Det | #P-hard w/o Det | Thm. 4.1&C.7 |
| | $(1-\alpha)^{-1}\log\int p^\alpha(\boldsymbol{x})q^{1-\alpha}(\boldsymbol{x})\,d\mathbf{X}, \alpha\in\mathbb{R}$ | Cmp, Det | #P-hard w/o Det | Thm. 4.1&C.7 |
| ITAKURA-SAITO DIV. | $\int[p(\boldsymbol{x})/q(\boldsymbol{x})-\log(p(\boldsymbol{x})/q(\boldsymbol{x}))-1]d\mathbf{X}$ | Cmp, Det | #P-hard w/o Det | Thm. C.8 |
| CAUCHY-SCHWARZ DIV. | $-\log\dfrac{\int p(\boldsymbol{x})q(\boldsymbol{x})d\mathbf{X}}{\sqrt{\int p^2(\boldsymbol{x})d\mathbf{X}\int q^2(\boldsymbol{x})d\mathbf{X}}}$ | Cmp | #P-hard w/o Cmp | Thm. C.9 |
| SQUARED LOSS | $\int(p(\boldsymbol{x})-q(\boldsymbol{x}))^2 d\mathbf{X}$ | Cmp | #P-hard w/o Cmp | Thm. C.10 |

We will show its practical utility by quickly coming up with tractability proofs as well as distilling efficient algorithms for several entropy and divergence queries that are largely used in ML. We will then discuss how our discovered tractable circuit classes subsume some previously known results in the literature and prove novel hardness results for when the structural properties of these circuits are unmet. Tab. 2 summarizes our results.

We now showcase how a short tractability proof can be easily distilled, using Rényi's $\alpha$-divergence[4] [40] as an example. Note that no tractable algorithm was available for it yet. A proof can be built by inferring the sufficient conditions to tractably compute each operator in the pipeline—starting from the last before the integral and proceeding backwards according to Tab. 1.

**Theorem 4.1** (Tractable alpha divergence). *The Rényi's $\alpha$-divergence between two distributions $p$ and $q$, defined as $(1-\alpha)^{-1}\log\int p^\alpha(\boldsymbol{x})q^{1-\alpha}(\boldsymbol{x})\,d\mathbf{X}$, can be computed exactly in $\mathcal{O}(|p|^\alpha |q|)$ time for $\alpha\in\mathbb{N}, \alpha>1$ if $p$ and $q$ are compatible and $q$ is deterministic, or in $\mathcal{O}(|p|\,|q|)$ time for $\alpha\in\mathbb{R}, \alpha\neq 1$ if $p$ and $q$ are both deterministic and compatible.*

*Proof.* A circuit pipeline for Rényi's $\alpha$-divergence involves first computing $r = p^\alpha$ and $s = q^{1-\alpha}$, then $t = r \cdot s$ and finally integrate it.[5] Therefore we require $t$ to be a smooth and decomposable circuit (Prop. 2.1), which in turn requires $r$ and $s$ to be compatible (Thm. 3.2). To conclude the proof, we need to compute two compatible circuits $r$ and $s$ in polytime, which can be done according to Thm. 3.5 or Thm. 3.3 depending on the value of $\alpha$. As these theorems state, $p^\alpha$ and $q^{1-\alpha}$ will be compatible with $p$ and $q$, respectively, with sizes $\mathcal{O}(|p|^\alpha)$ and $\mathcal{O}(|q|)$ for a natural power $\alpha$ or $\mathcal{O}(|p|)$ and $\mathcal{O}(|q|)$ for a real-valued $\alpha$. As such, $t$ could be computed in $\mathcal{O}(|p|^\alpha |q|)$ time for $\alpha\in\mathbb{N}$ or $\mathcal{O}(|p|\,|q|)$ for $\alpha\in\mathbb{R}$ (Thm. 3.2). $\square$

We leave the formal theorems and proofs for the other queries listed in Tab. 2 to Sec. C in the Appendix for space constraints. We remark again that our technique can be used beyond this query list and *can be applied to any complex query that involves a pipeline comprising the operations we discussed in Sec. 3 and culminating in an integration.*

**Shannon entropy** Smooth, decomposable and deterministic PCs enable the exact computation of Shannon entropy (Thm. C.2) and this tractability result translates to bounded-treewidth PGMs such as Chow-Liu trees and polytrees as they are special cases (Sec. 2). Our framework provides a more succinct tractability proof for the computation of Shannon entropy derived by Shih and Ermon [44], which we complete by demonstrating in Thm. C.2 that it is coNP-hard for non-deterministic PCs.

**Rényi entropy** For non-deterministic PCs we can employ the tractable computation of Rényi entropy of order $\alpha\in\mathbb{N}$ [40], which recovers Shannon Entropy for $\alpha\to 1$. As the logarithm is taken after integration of the power circuit, the tractability and hardness follow directly from those of the power operation (Thm. 3.3 and 3.5).

---

[4]Several alternative formulations of $\alpha$-divergences can be found in the literature such as Amari's [31] and Tsallis's [34] divergences. However, as they share the same core operations—real powers and products of circuits—our results easily extend to them as well.

[5]Note that all the operations outside integration are tractable, therefore we can skip them.

Table 3: ***Efficient algorithms for several query classes quickly distilled using our compositional atlas.*** Times in seconds to compute the Shannon entropy (ENT), cross-entropy (XENT), Kullback-Leibler divergence (KLD), Alpha divergence (AlphaDiv) for $\alpha = 1.5$, Rényi entropy (RényiEnt), and Cauchy-Schwarz divergence (CSDiv) over the circuits learned from 7 real-world datasets (complete results for 10 datasets in Tab. 5) by using algorithms distilled by our pipelines and comparing them to the highly-optimized implementations of the ENT [44] and KLD [28] algorithms available in Juice.jl [12]. No tractable implementation was available for XENT, AlphaDiv, RényiEnt, and CSDiv.

| DATASET | ENT | | KLD | | XENT | | ALPHADIV | | RÉNYIENT | | CSDIV | |
|---|---|---|---|---|---|---|---|---|---|---|---|---|
| | OURS | JUICE | OURS | JUICE | OURS | JUICE | OURS | JUICE | OURS | JUICE | OURS | JUICE |
| KDD | 0.157 | 0.001 | 3.154 | 0.790 | 2.180 | - | 0.885 | - | 0.016 | - | 1.136 | - |
| PLANTS | 0.679 | 0.005 | 3.983 | 3.909 | 3.739 | - | 1.160 | - | 0.088 | - | 1.572 | - |
| AUDIO | 0.406 | 0.003 | 2.736 | 1.681 | 1.873 | - | 0.537 | - | 0.029 | - | 0.771 | - |
| JESTER | 0.764 | 0.003 | 1.019 | 0.432 | 0.805 | - | 0.351 | - | 0.024 | - | 0.476 | - |
| NETFLIX | 0.106 | 0.002 | 0.352 | 0.175 | 0.264 | - | 0.100 | - | 0.017 | - | 0.201 | - |
| DNA | 4.365 | 0.027 | 64.664 | 220.377 | 52.997 | - | 15.609 | - | 0.255 | - | 22.901 | - |
| AD | 0.193 | 0.007 | 0.346 | 0.046 | 0.281 | - | 0.151 | - | 0.031 | - | 0.207 | - |

**Cross entropy** As hinted by the presence of a logarithm, the cross entropy is #P-hard to compute without determinism, even for compatible PCs (Thm. C.1). Nevertheless, given our atlas the cross entropy can be tractably computed in $\mathcal{O}(|p|\,|q|)$ if $p$ and $q$ are deterministic and compatible.

**Mutual information** Let a joint distribution $p(\mathbf{X}, \mathbf{Y})$ and its marginals $p(\mathbf{X})$ and $p(\mathbf{Y})$ be represented as PCs. Then the mutual information (MI) over these three PCs can be computed via a pipeline involving product, quotient, and log operators and it is tractable if all circuits are compatible and deterministic. On the other hand, if the marginal distributions cannot be represented as compact deterministic PCs, we prove it to be coNP-hard (Thm. C.3).

**Divergences** Liang and Van den Broeck [28] proposed an efficient algorithm to compute the KLD tailored for PSDDs. This has been the only tractable divergence available for PCs so far. We greatly extend this panorama with our atlas by introducing Rényi's $\alpha$-divergences which generalize several other divergences such as the KLD when $\alpha \to 1$, Hellinger's squared divergence when $\alpha = 2^{-1}$, and the $\mathcal{X}^2$-divergence when $\alpha = 2$ [16]. As Thm. 4.1 states, they are tractable for compatible and deterministic PCs, as is the Itakura-Saito divergence [53] (Thm. C.8). For non-deterministic PCs, we characterize the tractability of the squared loss and the Cauchy-Schwarz divergence [19]. The latter has applications in mixture models for approximate inference [46] and has been derived in closed-form only for mixtures of simple parametric forms like Gaussians [22], Weibull and Rayligh distributions [33]. Our results generalize them to deep mixture models [38].

**Expectation queries** Among other complex queries that can be abstracted into the general form of an expectation of a circuit $f$ w.r.t. a PC $p$, i.e., $\mathbb{E}_{\boldsymbol{x} \sim p(\mathbf{X})}\left[f(\boldsymbol{x})\right]$, there are the moments of distributions, such as means and variances. They can be efficiently computed for any smooth and decomposable PC, as $f$ is an omni-compatible circuit (Prop. D.1). This result generalizes the moment computation for simple models such as GMMs and HMMs as they can be encoded as smooth and decomposable PCs (Sec. 2). If $f$ is the indicator function of a logical formula, the expectation computes its probability w.r.t. the distribution $p$. Choi et al. [4] proposed an algorithm tailored to formulas $f$ over binary variables, encoded as SDDs [14] w.r.t. distributions that are PSDDs. We generalize this result to mixed continuous-discrete distributions encoded as structured-decomposable PCs that are not necessarily deterministic and to logical formulas in the language of satisfiability modulo theories [2] over linear arithmetics with univariate literals (Prop. D.2). Lastly, if $f$ encodes constraints over the output distribution of a deep network we retrieve the *semantic loss* [54]. If $f$ encodes a classifier or a regressor, then $\mathbb{E}_p[f]$ refers to computing its expected predictions w.r.t. $p$ [24]. Our results generalize the results reported in Van den Broeck et al. [47] such as computing the expectations of decision trees and their ensembles [25] (cf. Prop. D.3) as well as those of *deep regression circuits* [23].[6]

## 5   Experiments

We prototyped the tractable operators defined in Sec. 3 as subroutines in Julia in the Juice.jl framework [12] to showcase how our modular atlas can practically and quickly help implement tractable

---

[6]Despite the name, regression circuits do not conform to our definition of circuits in Def. 2.1. Nevertheless, we can translate them to our format in polytime as we illustrate in Alg. 8 in the Appendix.

algorithms for novel query classes. [7] To this extent, we distilled algorithms for the Shannon, Rényi, and cross entropies and for the KL, Alpha, and Cauchy-Schwarz divergences. We then ran them on deterministic and structured-decomposable circuits learned as in Dang et al. [11] from 20 publicly available real-world benchmark datasets [29, 48].

Tab. 3 shows the time taken to build and execute the pipelines on a subset of the datasets, while Tab. 4 and Tab. 5 in Sec. E in the Appendix report all the intermediate circuit sizes in their respective pipelines and times for all datasets. First, the intermediate circuits created in the pipeline do not blow up in size: as predicted by our theoretical analysis, the size of the logarithm circuit grows by a linear factor ($\sim$3–4x). Moreover, the size of the product circuit $p \cdot q$ is only slightly larger than $\max(|p|, |q|)$ when $p$ and $q$ are deterministic, much smaller than the theoretical bound of $\mathcal{O}(|p| |q|)$.

In terms of execution time, our algorithms run in less than a second for most circuits and peak at slightly more than one minute to compute a pipeline of the KLD, whose output circuit has more than 3 million edges on the DNA dataset (Tab. 4). A custom and highly optimized implementation of the KLD for PSDDs by Liang and Van den Broeck [28] runs up to ten times faster on smaller circuits but surprisingly takes $\sim$220 seconds for DNA, highlighting that our compositional atlas is a promising way to distill tractable algorithms. We emphasize that the aim of these experiments is not to distill the fastest algorithm for a query class, but to demonstrate that *our compositional framework empowers practitioners to quickly distill new tractable algorithms for queries that were not available before*, such as the Rényi entropy and $\alpha$ and Cauchy-Schwarz divergences.

## 6 Discussion and Conclusions

This work introduced a unified framework to reason about tractable model classes for complex queries composed of simpler operations. This rich atlas of operators can be used to solve many queries common in probabilistic ML and AI as well as novel inference scenarios.

Darwiche and Marquis [15] is the work most closely related to ours: they define operators over *logical circuits*, encoding Boolean functions as computational graphs with AND and OR gates, for which structural properties analogous to those discussed in Sec. 2 can be defined. Our results generalize their work on logical tractable operators such as disjunctions and conjunctions—the analogous to our (deterministic) sums and products—while also extending it to powers, logarithms and exponentials as well as complex queries such as divergences, which have no direct counterpart in the logical domain. Algorithms to tractably multiply two probabilistic models have been proposed for probabilistic decision graphs (PDGs) first [17] and PSDDs later [43]. Despite the different syntax, both model classes can be encoded as structured-decomposable and deterministic circuits in our language [8]. Historically, algorithms for the tractable product of PDGs and PSDDs relied on a special case of compatibility, when two structured-decomposable models exactly share the same hierarchical scope partitioning in terms of special graphical representations such as pseudo forests [17, 18] and vtrees [37]. They also implicitly entangle compatibility with determinism. As we showed in Thm. 3.2, compatibility is sufficient for tractable multiplication, which interestingly, has been noted in the context of logical circuits [37]. As discussed in the previous section, many algorithms tailored for PSDDs [4, 43, 23] can therefore be generalized to *non-deterministic* distributions in our framework.

Our property-driven analysis closes many open questions about the tractability and hardness of queries for several model classes that are special cases of circuits. At the same time, our hardness results might limit its general applicability by restricting tractable inference of some query classes to only certain inputs (Sec. 3). However, we see this as an opportunity: now that we can analyze which operators in a pipeline are tractable or not, we could substitute the latter by an approximate inference algorithm distilled from the pipeline. Lastly, we plan to extend our analysis to other queries involving not only integration but also maximization—that is, understanding what are the operators that make MAP inference over probabilistic circuits or optimization over general circuits tractable.

**Acknowledgements**    AV would like to thank Yujia Shen and Arthur Choi for insightful discussions about the product algorithm for PSDDs and Zhe Zeng for proofreading an initial version of this work. This work is partially supported by NSF grants #IIS-1943641, #IIS-1956441, #CCF-1837129, a Sloan Fellowship, and gifts from Intel and Facebook Research. The research of ST was partially supported by TAILOR, a project funded by EU Horizon 2020 research and innovation programme under GA No 952215.

---

[7]Code publicly available at `https://github.com/UCLA-StarAI/circuit-ops-atlas`.

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
