---

**Algorithm 1** SUPPORT($p$, cache)

---

1: **Input:** a smooth, deterministic, and decomposable circuit $p$ over variables $\mathbf{X}$ and a cache for memorization
2: **Output:** a smooth, deterministic, and decomposable circuit $s$ over $\mathbf{X}$ encoding $s(\boldsymbol{x}) = [\![\boldsymbol{x} \in \mathsf{supp}(p)]\!]$
3: **if** $p \in$ cache **then return** cache($p$)
4: **if** $p$ is an input unit **then** $s \leftarrow \text{INPUT}([\![\boldsymbol{x} \in \mathsf{supp}(p)]\!], \phi(p))$
5: **else if** $p$ is a sum unit **then** $s \leftarrow \text{SUM}(\{\text{SUPPORT}(p_i, \text{cache})\}_{i=1}^{|\mathsf{in}(p)|}, \{1\}_{i=1}^{|\mathsf{in}(p)|})$
6: **else if** $p$ is a product unit **then** $s \leftarrow \text{PRODUCT}(\{\text{SUPPORT}(p_i, \text{cache})\}_{i=1}^{|\mathsf{in}(p)|}|)$
7: cache($p$) $\leftarrow s$
8: **return** $s$

---

# A  Useful Sub-Routines

This section introduces the algorithmic construction of gadget circuits that will be adopted in our proofs of tractability as well as hardness. We start by introducing three primitive functions for constructing circuits—INPUT, SUM, and PRODUCT.

• INPUT($l_p, \phi(p)$) constructs an input unit $p$ that encodes a parameterized function $l_p$ over variables $\phi(p)$. For example, INPUT($[\![X = \text{True}]\!], X$) and INPUT($[\![X = \text{False}]\!], X$) represent the positive and negative literals of a Boolean variable $X$, respectively. On the other hand, INPUT($\mathcal{N}(\mu, \sigma), X$) defines a Gaussian pdf with mean $\mu$ and standard deviation $\sigma$ over variable $X$ as an input function.

• SUM($\{p_i\}_{i=1}^k, \{\theta_i\}_{i=1}^k$) constructs a sum unit that represents the weighted combination of $k$ circuit units $\{p_i\}_{i=1}^k$ encoded as an ordered set w.r.t. the correspondingly ordered weights $\{\theta_i\}_{i=1}^k$.

• PRODUCT($\{p_i\}_{i=1}^k$) builds a product unit that encodes the product of $k$ circuit units $\{p_i\}_{i=1}^k$.

## A.1  Support circuit of a deterministic circuit

Given a smooth, decomposable, and deterministic circuit $p(\mathbf{X})$, its support circuit $s(\mathbf{X})$ is a smooth, decomposable, and deterministic circuit that evaluates 1 iff the input $\boldsymbol{x}$ is in the support of $p$ (i.e., $\boldsymbol{x} \in \mathsf{supp}(p)$) and otherwise evaluates 0, as defined below.

**Definition A.1** (Support circuit). Let $p$ be a smooth, decomposable, and deterministic PC over variables $\mathbf{X}$. Its support circuit is the circuit $s$ that computes $s(\boldsymbol{x}) = [\![\boldsymbol{x} \in \mathsf{supp}(p)]\!]$, obtained by replacing every sum parameter of $p$ by 1 and every input distribution $l$ by the function $[\![\boldsymbol{x} \in \mathsf{supp}(l)]\!]$.

A construction algorithm for the support circuit is provided in Alg. 1. This algorithm will later be useful in defining some circuit operations such as the logarithm.

## A.2  Circuits encoding uniform distributions

We can build a deterministic and omni-compatible PC that encodes a (possibly unnormalized) uniform distribution over binary variables $\mathbf{X} = \{X_1, \dots, X_n\}$: i.e., $p(\boldsymbol{x}) = c$ for a constant $c \in \mathbb{R}_+$ for all $\boldsymbol{x} \in \mathsf{val}(\mathbf{X})$. Specifically, $p$ can be defined as a single sum unit with weight $c$ that receives input from a product unit over $n$ univariate input distribution units that always output 1 for all values $\mathsf{val}(X_i)$. This construction is summarized in Alg. 2. It is a key component in the algorithms for many tractable circuit transformations/queries as well as in several hardness proofs.

## A.3  A circuit representation of the **#3SAT** problem

We define a circuit representation of the #3SAT problem, following the construction in Khosravi et al. [23]. Specifically, we represent each instance in the #3SAT problem as two poly-sized structured-decomposable and deterministic circuits $p_\beta$ and $p_\gamma$, such that the partition function of their product equals the solution of the original #3SAT problem.

**#3SAT** is defined as follows: given a set of $n$ boolean variables $\mathbf{X} = \{X_1, \dots, X_n\}$ and a CNF that contains $m$ clauses $\{c_1, \dots, c_m\}$ (each clause contains exactly 3 literals), count the number of satisfiable worlds in $\mathsf{val}(\mathbf{X})$.

---

**Algorithm 2** UNIFORMCIRCUIT($\mathbf{X}, c$)

---

1: **Input:** a set of variables $\mathbf{X}$ and constant $c \in \mathbb{R}_+$.
2: **Output:** a deterministic and omni-compatible PC encoding an unnormalized uniform distribution over $\mathbf{X}$.
3: $n \leftarrow \{\}$
4: **for** $i = 1$ **to** $|\mathbf{X}|$ **do**
5:     $m \leftarrow \{\}$
6:     **for** $x_i$ **in** $\mathsf{val}(X_i)$ **do**
7:        $m \leftarrow m \cup \{\text{INPUT}(\llbracket X_i = x_i \rrbracket, X_i)\}$
8:     $n \leftarrow n \cup \{\text{SUM}(m, \{1\}_{j=1}^{|\mathsf{val}(X_i)|})\}$
9: **return** $\text{SUM}(\{\text{PRODUCT}(n)\}, \{c\})$

---

For every variable $X_i$ in clause $c_j$, we introduce an auxiliary variable $X_{ij}$. Intuitively, $\{X_{ij}\}_{j=1}^m$ are copies of the variable $X_i$, one for each clause. Therefore, for any $i$, $\{X_{ij}\}_{j=1}^m$ share the same value (i.e., true or false), which can be represented by the following formula $\beta$:

$$\beta \equiv \bigwedge_{i=1}^{n} (X_{i1} \Leftrightarrow X_{i2} \Leftrightarrow \cdots \Leftrightarrow X_{im}).$$

Then we can encode the original CNF in the following formula $\gamma$ by substituting $X_i$ with the respective $X_{ij}$ in each clause:

$$\gamma \equiv \bigwedge_{j=1}^{m} \bigvee_{i: X_i \in \phi(c_j)} l(X_{ij}),$$

where $\phi(c)$ denotes the variable scope of clause $c$, and $l(X_{ij})$ denotes the literal of $X_i$ in clause $c_j$. Since $\beta$ restricts the variables $\{X_{ij}\}_{j=1}^m$ to have the same value, the model count of $\beta \wedge \gamma$ is equal to the model count of the original CNF.

We are left to show that both $\beta$ and $\gamma$ can be compiled into a poly-sized structured-decomposable and deterministic circuit. We start from compiling $\beta$ into a circuit $p_\beta$. Note that for each $i$, $(X_{i1} \Leftrightarrow \cdots \Leftrightarrow X_{im})$ has exactly two satisfiable variable assignments (i.e., all true or all false), it can be compiled as a sum unit $a_i$ over two product units $b_{i1}$ and $b_{i2}$ (both weights of $a$ are set to 1), where $b_{i1}$ takes inputs from the positive literals $\{X_{i1}, \ldots, X_{im}\}$ and $b_{i2}$ from the negative literals $\{\neg X_{i1}, \ldots, \neg X_{im}\}$. Then $p_\beta$ is represented by a product unit over $\{a_1, \ldots, a_n\}$. Note that by definition this $p_\beta$ circuit is structured-decomposable and deterministic.

We proceed to compile $\gamma$ into a polysized structured-decomposable and deterministic circuit $p_\gamma$. Note that in #3SAT, each clause $c_j$ contains 3 literals. Therefore, for any $j \in \{1, \ldots, m\}$, $\bigvee_{X_i \in \phi(c_j)} l(X_{ij})$ has exactly 7 models w.r.t. the variable scope $\phi(c_j)$. Hence, we compile $\bigvee_{X_i \in \phi(c_j)} l(X_{ij})$ into a circuit $d_j$, which is a sum unit with 7 inputs $\{e_{j1}, \ldots, e_{j7}\}$. Each $e_{jh}$ is constructed as a product unit over variables $\{X_{1j}, \ldots, X_{nj}\}$ that represents the $h$-th model of clause $c_j$. More formally, we have $e_{jh} \leftarrow \text{PRODUCT}(\{g_{ijh}\}_{i=1}^n)$, where $g_{ijh}$ is a sum unit over literals $X_{ij}$ and $\neg X_{ij}$ (with both weights being 1) if $i \notin \phi(c_j)$ and otherwise $g_{ijh}$ is the literal unit corresponds to the $h$-th model of clause $c_j$. The circuit $p_\gamma$ representing the formula $\gamma$ is constructed by a product unit with inputs $\{d_j\}_{j=1}^m$. By construction this circuit is also structured-decomposable and deterministic.

## B   Circuit Operations

This section formally presents the tractability and hardness results w.r.t. circuit operations summarized in Tab. 1—sums, products, quotients, powers, logarithms, and exponentials. For each circuit operation, we provide both its proof of tractability by constructing a polytime algorithm given sufficient structural constraints and novel hardness results that identify necessary structural constraints for the operation to yield a decomposable circuit as output.

Throughout this paper, we will show hardness of operations to output a decomposable circuit by proving hardness of computing the partition function of the output of the operation. This follows from

the fact that we can smooth and integrate a decomposable circuit in polytime (Prop. 2.1), thereby making the former problem at least as hard as the latter.

For the tractability theorems, we will assume that the operation referenced by the theorem is tractable over input units of circuit or pairs of compatible input units. For example, for Thm. 3.2 we assume tractable product of input units sharing the same scope and for Thm. 3.5 we assume that the powers of the input units can be tractably represented as a single new unit. Note that this is generally easy to realize for simple parametric forms e.g., multivariate Gaussians and for univariate distributions, unless specified otherwise.

Moreover, in the following results, we will adopt a more general definition of compatibility that can be applied to circuits with different variable scopes, which is often useful in practice. Formally, consider two circuits $p$ and $q$ with variable scope $\mathbf{Z}$ and $\mathbf{Y}$. Analogous to Def. 2.5, we say that $p$ and $q$ are compatible over variables $\mathbf{X} = \mathbf{Z} \cap \mathbf{Y}$ if (1) they are smooth and decomposable and (2) any pair of product units $n \in p$ and $m \in q$ with the same overlapping scope with $\mathbf{X}$ can be rearranged into mutually compatible binary products. Note that since our tractability results hold for this extended definition of compatibility, they are also satisfied under Def. 2.5.

## B.1 Sum of Circuits

The hardness of the sum of two circuits to yield a deterministic circuit has been proven by Shen et al. [43] in the context of arithmetic circuits (ACs) [15]. ACs can be readily turned into circuits over binary variables according to our definition by translating their input parameters into sum parameters as done in Rooshenas and Lowd [41].

A sum of circuits will preserve decomposability and related properties as the next proposition details.

**Proposition B.1** (Closure of sum of circuits). *Let $p(\mathbf{Z})$ and $q(\mathbf{Y})$ be decomposable circuits. Then their sum circuit $s(\mathbf{Z} \cup \mathbf{Y}) = \theta_1 \cdot p(\mathbf{Z}) + \theta_2 \cdot q(\mathbf{Y})$ for two reals $\theta_1, \theta_2 \in \mathbb{R}$ is decomposable. If $p$ and $q$ are structured-decomposable and compatible, then $s$ is structured-decomposable and compatible with both $p$ and $q$. Lastly, if both inputs are also smooth, $s$ can be smoothed in polytime.*

*Proof.* If $p$ and $q$ are decomposable, $s$ is also decomposable by definition (no new product unit is introduced). If they are also structured-decomposable and compatible, $s$ would be structured-decomposable and compatible with $p$ and $q$ as well, as summation does not affect their hierarchical scope partitioning. Note that if one input is decomposable and the other omni-compatible, then $s$ would only be decomposable.

If $\mathbf{Z} = \mathbf{Y}$ then $s$ is smooth; otherwise we can smooth it in polytime [13, 45], by realizing the circuit

$$s(\boldsymbol{x}) = \theta_1 \cdot p(\boldsymbol{z}) \cdot [\![q(\boldsymbol{x}|_{\mathbf{Y} \setminus \mathbf{Z}}) \neq 0]\!] + \theta_2 \cdot q(\boldsymbol{y}) \cdot [\![p(\boldsymbol{x}|_{\mathbf{Z} \setminus \mathbf{Y}}) \neq 0]\!]$$

where $[\![q(\boldsymbol{x}|_{\mathbf{Y} \setminus \mathbf{Z}}) \neq 0]\!]$ (resp. $[\![p(\boldsymbol{x}|_{\mathbf{Z} \setminus \mathbf{Y}}) \neq 0]\!]$ ) can be encoded as an input distribution over variables $\mathbf{Y} \setminus \mathbf{Z}$ (resp. $\mathbf{Z} \setminus \mathbf{Y}$). Note that if the supports of $p(\mathbf{Z} \setminus \mathbf{Y})$ and $q(\mathbf{Y} \setminus \mathbf{Z})$ are not bounded, then integrals over them would be unbounded as well. $\square$

## B.2 Product of Circuits

**Theorem 3.1** (Hardness of product). Let $p$ and $q$ be two structured-decomposable and deterministic circuits over variables $\mathbf{X}$. Computing their product $m(\mathbf{X}) = p(\mathbf{X}) \cdot q(\mathbf{X})$ as a decomposable circuit is #P-hard.[8]

*Proof.* As noted earlier, we will prove hardness of computing the product by showing hardness of computing the partition function of a product of two circuits. In particular, let $p$ and $q$ be two structured-decomposable and deterministic circuits over binary variables $\mathbf{X}$. Then, computing the following quantity is #P-hard:

$$\sum_{\boldsymbol{x} \in \mathsf{val}(\mathbf{X})} p(\boldsymbol{x}) \cdot q(\boldsymbol{x}). \tag{MULPC}$$

---

[8]Note that this implies that product of decomposable circuits is also #P-hard, as decomposability is a weaker condition than structured-decomposability. The hardness results throughout this paper translate directly when input properties are relaxed.

The following proof is adapted from the proof of Thm. 2 in Khosravi et al. [23]. We reduce the #3SAT problem defined in Sec. A.3, which is known to be #P-hard, to MULPC. Recall that $p_\beta$ and $p_\gamma$, as constructed in Sec. A.3, are structured-decomposable and deterministic; additionally, the partition function of $p_\beta \cdot p_\gamma$ is the solution of the corresponding #3SAT problem. In other words, computing MULPC of two structured-decomposable and deterministic circuits $p_\beta$ and $p_\gamma$ exactly solves the original #3SAT problem. Therefore, computing the product of two structured-decomposable and deterministic circuits is #P-hard. $\qquad\square$

**Theorem 3.2** (Tractable product of circuits). Let $p(\mathbf{Z})$ and $q(\mathbf{Y})$ be two compatible circuits over variables $\mathbf{X} = \mathbf{Z} \cap \mathbf{Y}$. Then, computing their product $m(\mathbf{X}) = p(\mathbf{Z}) \cdot q(\mathbf{Y})$ as a decomposable circuit can be done in $\mathcal{O}(|p|\,|q|)$ time. If both $p$ and $q$ are also deterministic, then so is $m$, moreover if $p$ and $q$ are structured-decomposable then $m$ is compatible with $p$ (and $q$) over $\mathbf{X}$.

*Proof.* The proof proceeds by showing that computing the product of (i) two smooth and compatible sum units $p$ and $q$ and (ii) two smooth and compatible product units $p$ and $q$ given the product circuits w.r.t. pairs of child units from $p$ and $q$ (i.e., $\forall r \in \mathsf{in}(p)\, s \in \mathsf{in}(q), (r \cdot s)(\mathbf{X})$) takes time $\mathcal{O}(|\mathsf{in}(p)|\,|\mathsf{in}(q)|)$. Then, by recursion, the overall time complexity is $\mathcal{O}(|p|\,|q|)$. Alg. 3 illustrates the overall process in detail.

If $p$ and $q$ are two sum units defined as $p(\boldsymbol{x}) = \sum_{i \in \mathsf{in}(p)} \theta_i p_i(\boldsymbol{x})$ and $q(\boldsymbol{x}) = \sum_{j \in \mathsf{in}(q)} \theta'_j q_j(\boldsymbol{x})$, respectively. Then, their product $m(\boldsymbol{x})$ can be broken down to the weighted sum of $|\mathsf{in}(p)| \cdot |\mathsf{in}(q)|$ circuits that represent the products of pairs of their inputs:

$$m(\boldsymbol{x}) = \left( \sum_{i \in \mathsf{in}(p)} \theta_i p_i(\boldsymbol{x}) \right) \left( \sum_{j \in \mathsf{in}(q)} \theta'_j q_j(\boldsymbol{x}) \right) = \sum_{i \in \mathsf{in}(p)} \sum_{j \in \mathsf{in}(q)} \theta_i \theta'_j (p_i q_j)(\boldsymbol{x}).$$

Note that this Cartesian product of units is a deterministic sum unit if both $p$ and $q$ were deterministic sum units, as $\mathsf{supp}(p_i q_j) = \mathsf{supp}(p_i) \cap \mathsf{supp}(q_j)$ are disjoint for different $i, j$.

If $p$ and $q$ are two product units defined as $p(\mathbf{X}) = p_1(\mathbf{X}_1)p_2(\mathbf{X}_2)$ and $q(\mathbf{X}) = q_1(\mathbf{X}_1)q_2(\mathbf{X}_2)$, respectively. Then, their product $m(\boldsymbol{x})$ can be constructed recursively from the product of their inputs:

$$m(\boldsymbol{x}) = p_1(\boldsymbol{x}_1)p_2(\boldsymbol{x}_2) \cdot q_1(\boldsymbol{x}_1)q_2(\boldsymbol{x}_2) = p_1(\boldsymbol{x}_1)q_1(\boldsymbol{x}_1) \cdot p_2(\boldsymbol{x}_2)q_2(\boldsymbol{x}_2) = (p_1 q_1)(\boldsymbol{x}_1) \cdot (p_2 q_2)(\boldsymbol{x}_2).$$

Note that by this construction $m$ retains the same scope partitioning of $p$ and $q$, hence if they were structured-decomposable, $m$ will be structured-decomposable and compatible with $p$ and $q$. $\qquad\square$

Possessing additional structural constrains can lead to sparser output circuits as well as efficient algorithms to construct them. First, if one among $p$ and $q$ is omni-compatible, it suffices that the other is just decomposable to obtain a tractable product, whose size this time is going to be linear in the size of the decomposable circuit.

**Corollary B.1.** *Let $p$ be a smooth and decomposable circuit over $\mathbf{X}$ and $q$ an omni-compatible circuit over $\mathbf{X}$ comprising a sum unit with $k$ inputs, hence its size is $k\,|\mathbf{X}|$. Then, $m(\mathbf{X}) = p(\mathbf{X})q(\mathbf{X})$ is a smooth and decomposable circuit constructed in $\mathcal{O}(k\,|p|)$ time.*

Second, if $p$ and $q$ have inputs with restricted supports, their product is going to be sparse, i.e., only a subset of their inputs is going to yield a circuit that does not constantly output zero. Note that in Alg. 3 we can check in polytime if the supports of two units to be multiplied are overlapping by a depth-first search (realized with a Boolean indicator $s$ in Alg. 3), thanks to decomposability. Therefore, for two compatible sum units $p$ and $q$ we will effectively build a number of units that is

$$\mathcal{O}(|\{(p_i, q_j)|p_i \in \mathsf{in}(p), q_i \in \mathsf{in}(q), \mathsf{supp}(p_i) \cap \mathsf{supp}(q_j) \neq \emptyset\}|).$$

In practice, this sparsifying effect will be more prominent when both $p$ and $q$ are deterministic. This is because having disjoint supports is required for deterministic circuits. This "decimation" of product units will be maximum if $p$ and $q$ partition the support in the very same way, for instance when we have $p = q$, i.e., we are multiplying one circuit with itself, or we are dealing with a logarithmic circuit (cf. Sec. B.5). In such a case, we can omit the depth-first check for overlapping supports of the product units participating in the product of a sum unit. If both $p$ and $q$ have an identifier for their supports, we can simply check for equality of their identifiers. This property and algorithmic insight will be key when computing powers of a deterministic circuit and its entropies (cf. Sec. C.2), as it would suffice the input circuit $p$ to be decomposable (cf. Sec. 3) to obtain a linear time complexity.

**Algorithm 3** MULTIPLY($p, q$, cache)

---

1: **Input:** two circuits $p(\mathbf{Z})$ and $q(\mathbf{Y})$ that are compatible over $\mathbf{X} = \mathbf{Z} \cap \mathbf{Y}$ and a cache for memoization
2: **Output:** their product circuit $m(\mathbf{Z} \cup \mathbf{Y}) = p(\mathbf{Z})q(\mathbf{Y})$
3: **if** $(p, q) \in$ cache **then return** cache$(p, q)$
4: **if** $\phi(p) \cap \phi(q) = \emptyset$ **then**
5:     $m \leftarrow$ PRODUCT$(\{p, q\}); \; s \leftarrow$ True
6: **else if** $p, q$ are input units **then**
7:     $m \leftarrow$ INPUT$(p(\mathbf{Z}) \cdot q(\mathbf{Y}), \mathbf{Z} \cup \mathbf{Y})$
8:     $s \leftarrow [\![ \text{supp}(p(\mathbf{X})) \cap \text{supp}(q(\mathbf{X})) \neq \emptyset ]\!]$
9: **else if** $p$ is an input unit **then**
10:     $n \leftarrow \{\}; s \leftarrow$ False $// q(\mathbf{Y}) = \sum_j \theta'_j q_j(\mathbf{Y})$
11:     **for** $j = 1$ **to** $|\text{in}(q)|$ **do**
12:         $n', s' \leftarrow$ MULTIPLY$(p, q_j,$ cache$)$
13:         $n \leftarrow n \cup \{n'\}; \; s \leftarrow s \vee s'$
14:     **if** $s$ **then** $m \leftarrow$ SUM$(n, \{\theta'_j\}_{j=1}^{|\text{in}(q)|})$ **else** $m \leftarrow null$
15: **else if** $q$ is an input unit **then**
16:     $n \leftarrow \{\}; s \leftarrow$ False $// p(\mathbf{Z}) = \sum_i \theta_i p_i(\mathbf{Z})$
17:     **for** $i = 1$ **to** $|\text{in}(p)|$ **do**
18:         $n', s' \leftarrow$ MULTIPLY$(p_i, q,$ cache$)$
19:         $n \leftarrow n \cup \{n'\}; \; s \leftarrow s \vee s'$
20:     **if** $s$ **then** $m \leftarrow$ SUM$(n, \{\theta_i\}_{i=1}^{|\text{in}(p)|})$ **else** $m \leftarrow null$
21: **else if** $p, q$ are product units **then**
22:     $n \leftarrow \{\}; s \leftarrow$ True
23:     $\{p_i, q_i\}_{i=1}^k \leftarrow$ sortPairsByScope$(p, q, \mathbf{X})$
24:     **for** $i = 1$ **to** $k$ **do**
25:         $n', s' \leftarrow$ MULTIPLY$(p_i, q_i,$ cache$)$
26:         $n \leftarrow n \cup \{n'\}; \; s \leftarrow s \wedge s'$
27:     **if** $s$ **then** $m \leftarrow$ PRODUCT$(n)$ **else** $m \leftarrow null$
28: **else if** $p, q$ are sum units **then**
29:     $n \leftarrow \{\}; \; w \leftarrow \{\}; \; s \leftarrow$ False
30:     **for** $i = 1$ **to** $|\text{in}(p)|, j = 1$ **to** $|\text{in}(q)|$ **do**
31:         $n', s' \leftarrow$ MULTIPLY$(p_i, q_j,$ cache$)$
32:         $n \leftarrow n \cup n'; w \leftarrow w \cup \{\theta_i \theta'_j\}; s \leftarrow s \vee s'$
33:     **if** $s$ **then** $m \leftarrow$ SUM$(n, w)$ **else** $m \leftarrow null$
34: cache$(p, q) \leftarrow (m, s)$
35: **return** $m, s$

---

## B.3 Power Function of Circuits

**Theorem 3.3** (Natural powers). *If $p$ is a structured-decomposable circuit, then for any $\alpha \in \mathbb{N}$, its power can be represented as a structured-decomposable circuit in $\mathcal{O}(|p|^\alpha)$ time. Otherwise, if $p$ is only smooth and decomposable, then computing $p^\alpha(\mathbf{X})$ as a decomposable circuit is #P-hard.*

*Proof.* The proof for tractability easily follows by directly applying the product operation repeatedly.

We prove hardness for the special case of discrete variables, and by showing the hardness of computing the partition function of $p^2(\mathbf{X})$. In particular, let $\mathbf{X}$ be a collection of binary variables and let $p$ be a smooth and decomposable circuit over $\mathbf{X}$, then computing the quantity

$$\sum_{\boldsymbol{x} \in \text{val}(\mathbf{X})} p^2(\boldsymbol{x}) \tag{POW2PC}$$

is #P-hard.

The proof builds a reduction from the #3SAT problem, which is known to be #P-hard. We employ the same setting of Sec. A.3, where a CNF over $n$ Boolean variables $\mathbf{X} = \{X_1, \ldots, X_n\}$ and containing

---

**Algorithm 4** SORTPAIRSBYSCOPE($p, q, \mathbf{X}$)

---

1: **Input:** two decomposable and compatible product units $p$ and $q$, and a variable scope $\mathbf{X}$.
2: **Output:** Pairs of compatible sum units $\{(p_i, q_i)\}_{i=1}^k$.
3: children_$p \leftarrow \{p_i\}_{i=1}^{|\text{in}(p)|}$,    children_$q \leftarrow \{q_i\}_{i=1}^{|\text{in}(q)|}$
4: pairs $\leftarrow \{\}$. // "pairs" stores circuit pairs with matched scope.
5: cmp_$p \leftarrow \{\{\}\}_{i=1}^{|\text{in}(p)|}$,    cmp_$q \leftarrow \{\{\}\}_{j=1}^{|\text{in}(q)|}$.
   // cmp_$p[i]$ (resp. cmp_$q[j]$) stores the children of $q$ (resp. $p$) whose scopes are subsets of $p_i$'s (resp. $q_j$'s) scope.
6: **for** i = 1 **to** $|\text{in}(p)|$ **do**
7:   **for** j = 1 **to** $|\text{in}(q)|$ **do**
8:     **if** $\phi(p_i) \cap \mathbf{X} = \phi(q_j) \cap \mathbf{X}$ **then**
9:       pairs.$append((p_i, q_j))$
10:       children_$p.pop(p_i)$,    children_$q.pop(q_j)$
11:     **else if** $\phi(p_i) \cap \mathbf{X} \subset \phi(q_j) \cap \mathbf{X}$ **then**
12:       cmp_$q[j].append(p_i)$
13:       children_$p.pop(p_i)$,    children_$q.pop(q_j)$
14:     **else if** $\phi(q_j) \cap \mathbf{X} \subset \phi(p_i) \cap \mathbf{X}$ **then**
15:       cmp_$p[i].append(q_j)$
16:       children_$p.pop(p_i)$,    children_$q.pop(q_j)$
17: **for** $i = 1$ **to** $|\text{in}(p)|$ **do**
18:   **if** $len(\text{cmp}\_p[i]) \neq 0$ **then**
19:     $s \leftarrow$ SUM($\{$PRODUCT(cmp_$p[i]$)$\}, \{1\}$)
20:     pairs.$append((p_i, s))$
21: **for** $j = 1$ **to** $|\text{in}(q)|$ **do**
22:   **if** $len(\text{cmp}\_q[j]) \neq 0$ **then**
23:     $r \leftarrow$ SUM($\{$PRODUCT(cmp_$q[j]$)$\}, \{1\}$)
24:     pairs.$append((r, q_j))$
25: **for** $r, s$ **in** $zip(\text{children}\_p, \text{children}\_q)$ **do**
26:   pairs.$append((r, s))$
27: **if** $len(\text{children}\_p) > len(\text{children}\_q)$ **then**
28:   **for** $i = len(\text{children}\_q) + 1$ **to** $len(\text{children}\_p)$ **do**
29:     pairs.$append((\text{children}\_p[i], \text{children}\_q[1]))$
30: **else if** $len(\text{children}\_p) < len(\text{children}\_q)$ **then**
31:   **for** $j = len(\text{children}\_p) + 1$ **to** $len(\text{children}\_q)$ **do**
32:     pairs.$append((\text{children}\_p[1], \text{children}\_q[j]))$
33: **return** $pairs$

---

$m$ clauses $\{c_1, \ldots, c_m\}$, each with exactly 3 literals, is encoded into two structured-decomposable and deterministic circuits $p_\beta$ and $p_\gamma$ over variables $\hat{\mathbf{X}} = \{X_{11}, \ldots, X_{1m}, \ldots, X_{n1}, \ldots, X_{nm}\}$.

Then, we construct circuit $p_\alpha$ as the sum of $p_\beta$ and $p_\gamma$, i.e., $p_\alpha(\hat{\boldsymbol{x}}) := p_\beta(\hat{\boldsymbol{x}}) + p_\gamma(\hat{\boldsymbol{x}})$. By definition $p_\alpha$ is smooth and decomposable, but not structured-decomposable. We proceed to show that if we can represent $p_\alpha^2(\hat{\boldsymbol{x}})$ as a smooth and decomposable circuit in polytime, we could solve POW2PC and hence #3SAT. That would mean that computing POW2PC is #P-hard.

By definition, $p_\alpha^2(\hat{\boldsymbol{x}}) = (p_\beta(\hat{\boldsymbol{x}}) + p_\gamma(\hat{\boldsymbol{x}}))^2 = p_\beta^2(\hat{\boldsymbol{x}}) + p_\gamma^2(\hat{\boldsymbol{x}}) + 2p_\beta(\hat{\boldsymbol{x}}) \cdot p_\gamma(\hat{\boldsymbol{x}})$, and hence

$$\sum_{\hat{\boldsymbol{x}} \in \text{val}(\hat{\mathbf{X}})} p_\alpha^2(\hat{\boldsymbol{x}}) = \sum_{\hat{\boldsymbol{x}} \in \text{val}(\hat{\mathbf{X}})} p_\beta^2(\hat{\boldsymbol{x}}) + \sum_{\hat{\boldsymbol{x}} \in \text{val}(\hat{\mathbf{X}})} p_\gamma^2(\hat{\boldsymbol{x}}) + \sum_{\hat{\boldsymbol{x}} \in \text{val}(\hat{\mathbf{X}})} p_\beta(\hat{\boldsymbol{x}}) \cdot p_\gamma(\hat{\boldsymbol{x}}).$$

Since $p_\beta$ and $p_\gamma$ are both structured-decomposable and deterministic the first two summations over the squared circuits can be computed in time $\mathcal{O}(|p_\beta| + |p_\gamma|)$ (see Thm. 3.5). It follows that if we could efficiently solve POW2PC we could then solve the that third summation, i.e., $\sum_{\hat{\boldsymbol{x}} \in \text{val}(\hat{\mathbf{X}})} p_\beta(\hat{\boldsymbol{x}}) \cdot p_\gamma(\hat{\boldsymbol{x}})$. However, since such a summation is the instance of MULPC between $p_\beta$ and $p_\gamma$ reduced from #3SAT (see Thm. 3.1), it would mean that we could solve #3SAT. We can conclude that computing POW2PC is #P-hard. $\qquad \square$

**Theorem B.1** (Hardness of natural power of a structured-decomposable circuit). *Let $p$ be a structured-decomposable circuit over variables $\mathbf{X}$. Let $k$ be a natural number. Then there is no polynomial $f(x, y)$ such that the power $p^k$ can be computed in $\mathcal{O}(f(|p|, k))$ time unless P=NP.*

*Proof.* We construct the proof by showing that for a structured-decomposable circuit $p$, if we could compute

$$\sum_{\boldsymbol{x} \in \mathsf{val}(\mathbf{X})} p^k(\boldsymbol{x}). \tag{POWkPC}$$

in $\mathcal{O}(f(|p|, k))$ time, then we could solve the 3SAT problem in polytime, which is known to be NP-hard.

The 3SAT problem is defined as follows: given a set of $n$ Boolean variables $\mathbf{X} = \{X_1, \ldots, X_n\}$ and a CNF that contains $m$ clauses $\{c_1, \ldots, c_m\}$, each one containing exactly 3 literals, determine whether there exists a satisfiable configuration in $\mathsf{val}(\mathbf{X})$.

We start by constructing $m$ gadget circuits $\{d_j\}_{j=1}^m$ for the $m$ clauses such that $d_j(\boldsymbol{x})$ evaluates to $\frac{1}{m}$ iff $\boldsymbol{x}$ satisfies $c_j$ and otherwise evaluates to 0, respectively.

Since each clause $c_j$ contains exactly 3 literals, it comprises exactly 7 models w.r.t. the variables appearing in it, i.e., its scope $\phi(c_j)$. Therefore, following a similar construction in Sec. A.3, we can compile $d_j$ as a weighted sum of 7 circuits that represent the 7 models of $c_j$, respectively. By choosing all weights of $d_j$ as $\frac{1}{m}$, the circuit $d_j$ outputs $\frac{1}{m}$ iff $c_j$ is satisfied; otherwise it outputs 0.

The gadget circuits $\{d_j\}_{j=1}^m$ are then summed together to represent a circuit $p$. That is, $p = \mathrm{SUM}(\{d_j\}_{j=1}^m, \{1\}_{j=1}^m)$. In the following, we complete the proof by showing that if the power circuit $p^k$ (we will pick later $k = \lceil \max(m, n)^2 \cdot \log 2 \rceil$) can be computed in $\mathcal{O}(f(|p|, k))$ time, then the corresponding 3SAT problem can be solved in $\mathcal{O}(f(|p|, k))$ time.

If the original CNF is satisfiable, then there exists at least 1 world such that all clauses are satisfied. In this case, all circuits in $\{d_j\}_{j=1}^m$ will evaluate $\frac{1}{m}$. Since $p$ is the sum of the circuits $\{d_j\}_{j=1}^m$, it will evaluate 1 for any world that satisfies the CNF. We obtain the bound

$$\sum_{\boldsymbol{x} \in \mathsf{val}(\mathbf{X})} p^k(\boldsymbol{x}) > m \cdot \frac{1}{m} = 1.$$

In contrast, if the CNF is unsatisfiable, each variable assignment $\boldsymbol{x} \in \mathsf{val}(\mathbf{X})$ satisfies at most $m - 1$ clauses, so the circuit $p$ will output at most $\frac{m-1}{m}$. Therefore , we retrieve the following bound

$$\sum_{\boldsymbol{x} \in \mathsf{val}(\mathbf{X})} p^k(\boldsymbol{x}) \leq 2^n \left( \frac{m-1}{m} \right)^k.$$

Then, we can retrieve a value for $k$ to separate the two bounds as follows.

$$2^n \left( \frac{m-1}{m} \right)^k < 1 \iff k > \frac{\log(2^{-n})}{\log \frac{m-1}{m}} \iff k > \frac{n \log 2}{\log(m) - \log(m-1)} \overset{(a)}{\iff} k > m \cdot n \cdot \log 2,$$

where $(a)$ follows the fact that $\log\left(\frac{m}{m-1}\right) \leq \frac{1}{m-1}$. Let $l = \max(m, n)$. If we choose $k = \lceil l^2 \cdot \log 2 \rceil$, then we can separate the two bounds above.

Therefore, if there exists a polynomial $f(x, y)$ such that the power $p^k$ ($k = \lceil l^2 \cdot \log 2 \rceil$) can be computed in $\mathcal{O}(f(|p|, k))$ time, then we can solve 3SAT in $\mathcal{O}(f(|p|, k))$ time since the CNF is satisfiable iff $\sum_{\boldsymbol{x} \in \mathsf{val}(\mathbf{X})} p^k(\boldsymbol{x}) > 1$, which is impossible unless P=NP. □

**Theorem 3.4** (Hardness of reciprocal of a circuit). Let $p$ be a smooth and decomposable circuit over variables $\mathbf{X}$. Then computing $p^{-1}(\mathbf{X})\big|_{\mathsf{supp}(p)}$ as a decomposable circuit is #P-hard, even if $p$ is structured-decomposable.

*Proof.* We prove it for the case of PCs over discrete variables. We will prove hardness of computing the reciprocal by showing hardness of computing the partition of the reciprocal of a circuit. In

particular, let $\mathbf{X} = \{X_1, \ldots, X_n\}$ be a collection of binary variables and let $p$ be a smooth and decomposable PC over $\mathbf{X}$, then computing the quantity

$$\sum_{\boldsymbol{x} \in \mathsf{val}(\mathbf{X})} \frac{1}{p(\boldsymbol{x})} \tag{INVPC}$$

is #P-hard.

Proof is by reduction from the EXPLR problem as defined in Thm. B.2. Similarly to Thm. B.2, the reduction is built by constructing a smooth and decomposable unnormalized circuit $p(x) = 2^n \cdot 1 + 2^n e^{-(w_0 + \sum_i w_i x_i)}$. The circuit $p$ comprises a sum unit over two sub-circuits. The first is a uniform (unnormalized) distribution over $\mathbf{X}$ defined as a product unit over $n$ univariate input distribution units that always output 1 for all values $\mathsf{val}(X_i)$ (see Sec. A.2 for a construction algorithm). The second is an exponential of a linear circuit (Alg. 7) and encodes $e^{-(w_0 + \sum_i w_i x_i)}$ via a product unit over $n$ univariate input distributions, where one of them encodes $e^{-w_0 - w_1 x_1}$ and the rest $e^{-w_j x_j}$ for $j = 2, \ldots, n$. Both sub-circuits participates in the sum with parameters $2^n$.

The size of the constructed circuit is linear in $n$, and INVPC of this circuit corresponds to the solution of the EXPLR problem. If you can represent the reciprocal of this circuit as a decomposable circuit, you can compute its marginals (including the partition function) which solves INVPC and hence EXPLR. Furthermore, the circuit is also omni-compatible because mixture of fully-factorized distributions. $\qquad\square$

**Theorem 3.5** (Tractable real power of a deterministic circuit)**.** Let $p$ be a smooth, decomposable, and deterministic circuit over variables $\mathbf{X}$. Then, for any real number $\alpha \in \mathbb{R}$, its restricted power, defined as $a(\boldsymbol{x})|_{\mathsf{supp}(p)} = p^\alpha(\boldsymbol{x})[\![\boldsymbol{x} \in \mathsf{supp}(p)]\!]$ can be represented as a smooth, decomposable, and deterministic circuit over variables $\mathbf{X}$ in $\mathcal{O}(|p|)$ time. Moreover, if $p$ is structured-decomposable, then $a$ is structured-decomposable as well.

*Proof.* The proof proceeds by construction and recursively builds $a(\boldsymbol{x})|_{\mathsf{supp}(p)}$. As the base case, we can assume to compute the restricted $\alpha$-power of the input units of $p$ and represent it as a single new unit. When we encounter a deterministic sum unit, the power will decompose into the sum of the powers of its inputs. Specifically, let $p$ be a sum unit: $p(\mathbf{X}) = \sum_{i \in \mathsf{in}(p)} \theta_i p_i(\mathbf{X})$. Then, its restricted real power circuit $a(\boldsymbol{x})|_{\mathsf{supp}(p)}$ can be expressed as

$$a(\boldsymbol{x})|_{\mathsf{supp}(p)} = \left( \sum_{i \in \mathsf{in}(p)} \theta_i p_i(\boldsymbol{x}) \right)^\alpha [\![\boldsymbol{x} \in \mathsf{supp}(p)]\!] = \sum_{i \in \mathsf{in}(p)} \theta_i^\alpha \big( p_i(\boldsymbol{x}) \big)^\alpha [\![\boldsymbol{x} \in \mathsf{supp}(p_i)]\!].$$

Note that this construction is possible because only one input of $p$ will be non-zero for any input (determinism). As such, the power circuit is retaining the same structure of the original sum unit.

Next, for a decomposable product unit, its power will be the product of the powers of its inputs. Specifically, let $p$ be a product unit: $p(\mathbf{X}) = p_1(\mathbf{X}_1) \cdot p_2(\mathbf{X}_2)$. Then, its restricted real power circuit $a(\boldsymbol{x})|_{\mathsf{supp}(p)}$ can be expressed as

$$a(\boldsymbol{x})|_{\mathsf{supp}(p)} = \big( p_1(\boldsymbol{x}_1) \cdot p_2(\boldsymbol{x}_2) \big)^\alpha [\![\boldsymbol{x} \in \mathsf{supp}(p)]\!]$$
$$= \big( p_1(\boldsymbol{x}_1) \big)^\alpha [\![\boldsymbol{x} \in \mathsf{supp}(p_1)]\!] \cdot \big( p_2(\boldsymbol{x}_2) \big)^\alpha [\![\boldsymbol{x} \in \mathsf{supp}(p_2)]\!].$$

Note that even this construction preserves the structure of $p$ and hence its scope partitioning is retained throughout the whole algorithm. Hence, if $p$ were also structured-decomposable, then $a$ would be structured-decomposable. Alg. 5 illustrates the whole algorithm in detail. $\qquad\square$

## B.4  Quotient of Circuits

**Theorem B.2** (Hardness of quotient of two circuits)**.** *Let $p$ and $q$ be two smooth and decomposable circuits over variables $\mathbf{X}$, and let $q(\boldsymbol{x}) \neq 0$ for every $\boldsymbol{x} \in \mathsf{val}(\mathbf{X})$. Then, computing their quotient $p(\mathbf{X})/q(\mathbf{X})$ as a decomposable circuit is #P-hard, even if they are compatible.*

**Algorithm 5** POWER($p, \alpha$, cache)
___
1: **Input:** a smooth, deterministic and decomposable circuit $p(\mathbf{X})$, a scalar $\alpha \in \mathbb{R}$, and a cache for memoization
2: **Output:** a smooth, deterministic and decomposable circuit $a(\mathbf{X})$ encoding $p^\alpha(\mathbf{X})|_{\mathsf{supp}(p)}$
3: **if** $p \in$ cache **then return** cache($p$)
4: **if** $p$ is an input unit **then** $a \leftarrow$ INPUT($p^\alpha(\mathbf{X})|_{\mathsf{supp}(p)}, \phi(p)$)
5: **else if** $p$ is a sum unit **then** $a \leftarrow$ SUM($\{$POWER($p_i, \alpha$, cache)$\}_{i=1}^{|\mathsf{in}(p)|}, \{\theta_i^\alpha\}_{i=1}^{|\mathsf{in}(p)|}$)
6: **else if** $p$ is a product unit **then** $a \leftarrow$ PRODUCT($\{$POWER($p_i, \alpha, ,$ cache)$\}_{i=1}^{|\mathsf{in}(p)|}$)
7: cache($p$) $\leftarrow a$
8: **return** $a$
___

*Proof.* This result follows from Thm. 3.4 by noting that computing the reciprocal of a circuit is a special case of computing the quotient of two circuits. In particular, let $p$ be an omni-compatible circuit representing the constant function 1 over variables $\mathbf{X}$, constructed as in Sec. A.2. Then computing the reciprocal of a structured-decomposable circuit $q$ as a decomposable circuit reduces to computing the quotient $p/q$. $\qquad\square$

**Theorem B.3** (Tractable restricted quotient of two circuits)**.** *Let $p$ and $q$ be two compatible circuits over variables $\mathbf{X}$, and let $q$ be also deterministic. Then, their quotient restricted to $\mathsf{supp}(q)$ can be represented as a circuit compatible with $p$ (and $q$) over variables $\mathbf{X}$ in $\mathcal{O}(|p|\,|q|)$ time. Moreover, if $p$ is also deterministic, then the quotient circuit is deterministic as well.*

*Proof.* We know from Thm. 3.5 that we can obtain the reciprocal circuit $q^{-1}$ that is also compatible with $q$ (and by extension $p$) in $\mathcal{O}(|q|)$ time. Then we can multiply $p$ and $q^{-1}$ in $\mathcal{O}(|p|\,|q|)$ time using Thm. 3.2 to compute their quotient circuit that is still compatible with $p$ and $q$. If $p$ is also deterministic, then we are multiplying two deterministic circuits and therefore their product circuit is deterministic (Thm. 3.2). $\qquad\square$

## B.5 Logarithm of a PC

**Theorem 3.6** (Logarithms)**.** *(Tractability)* Let $p$ be a smooth, deterministic and decomposable PC over variables $\mathbf{X}$. Then its logarithm circuit, restricted to the support of $p$ and defined as

$$l(\boldsymbol{x})|_{\mathsf{supp}(p)} = \begin{cases} \log p(\boldsymbol{x}) & \text{if } \boldsymbol{x} \in \mathsf{supp}(p) \\ 0 & \text{otherwise} \end{cases}$$

for every $\boldsymbol{x} \in \mathsf{val}(\mathbf{X})$ can be represented as a smooth and decomposable circuit that shares the scope partitioning of $p$ in $\mathcal{O}(|p|)$ time. *(Hardness)* Otherwise, if $p$ is a smooth and decomposable PC, then computing its logarithm circuit $l(\mathbf{X}) := \log p(\mathbf{X})$ as a decomposable circuit is #P-hard, even if $p$ is structured-decomposable.

We will provide the proofs for tractability and hardness separately below.

*Proof of tractability.* The proof proceeds by recursively constructing $l(\boldsymbol{x})|_{\mathsf{supp}(p)}$. In the base case, we assume computing the logarithm of an input unit can be done in $\mathcal{O}(1)$ time. When we encounter a deterministic sum unit $p(\boldsymbol{x}) = \sum_{i\in|\mathsf{in}(p)|} \theta_i p_i(\boldsymbol{x})$, its logarithm circuit consists of the sum of (i) the logarithm circuits of its child units and (ii) the support circuits of its children weighted by their respective weights $\{\theta_i\}_{i=1}^{|\mathsf{in}(p)|}$:

$$l(\boldsymbol{x})|_{\mathsf{supp}(\boldsymbol{x})} = \log\left(\sum_{i\in\mathsf{in}(p)} \theta_i p_i(\boldsymbol{x})\right) \cdot [\![\boldsymbol{x} \in \mathsf{supp}(p)]\!] = \sum_{i\in|\mathsf{in}(p)|} \log\left(\theta_i p_i(\boldsymbol{x})\right)[\![\boldsymbol{x} \in \mathsf{supp}(p_i)]\!]$$

$$= \sum_{i\in|\mathsf{in}(p)|} \log \theta_i [\![\boldsymbol{x} \in \mathsf{supp}(p_i)]\!] + \sum_{i\in|\mathsf{in}(p)|} l_i(\boldsymbol{x})|_{\mathsf{supp}(p_i)}.$$

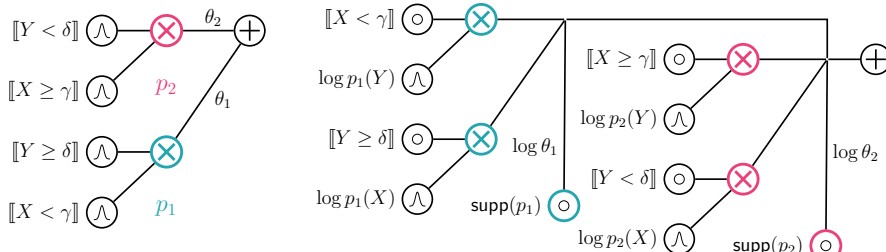

Figure 3: Building the logarithmic circuit (right) for a deterministic PC (left) whose input units are labeled by their supports. A single sum unit is introduced over smoothed product units and additional dummy input units which share the same support across circuits if they have the same color.

For a smooth, decomposable, and deterministic product unit $p(\boldsymbol{x}) = p_1(\boldsymbol{x})p_2(\boldsymbol{x})$, its logarithm circuit can be decomposed as sum of the logarithm circuits of its child units:

$$l(\boldsymbol{x})|_{\mathsf{supp}(\boldsymbol{x})} = \log\left(p_1(\boldsymbol{x}_1)p_2(\boldsymbol{x}_2)\right) \cdot [\![\boldsymbol{x} \in \mathsf{supp}(p)]\!]$$
$$= \log p_1(\boldsymbol{x}_1)[\![\boldsymbol{x} \in \mathsf{supp}(p)]\!] + \log p_2(\boldsymbol{x}_2)[\![\boldsymbol{x} \in \mathsf{supp}(p)]\!]$$
$$= \log p_1(\boldsymbol{x}_1)[\![\boldsymbol{x}_1 \in \mathsf{supp}(p_1)]\!][\![\boldsymbol{x}_2 \in \mathsf{supp}(p_2)]\!] + \log p_2(\boldsymbol{x}_2)[\![\boldsymbol{x}_2 \in \mathsf{supp}(p_2)]\!][\![\boldsymbol{x}_1 \in \mathsf{supp}(p_1)]\!]$$
$$= l(\boldsymbol{x}_1)|_{\mathsf{supp}(p_1)} [\![\boldsymbol{x}_2 \in \mathsf{supp}(p_2)]\!] + l(\boldsymbol{x}_2)|_{\mathsf{supp}(p_2)} [\![\boldsymbol{x}_1 \in \mathsf{supp}(p_1)]\!].$$

Note that in both case, the support circuits (e.g., $[\![\boldsymbol{x} \in \mathsf{supp}(p)]\!]$) are used to enforce smoothness in the output circuit. Alg. 6 illustrates the whole algorithm in detail, showing that the construction of these support circuits can be done in linear time by caching intermediate sub-circuits while calling Alg. 1. Furthermore, the newly introduced product units, i.e., $l(\boldsymbol{x}_1)|_{\mathsf{supp}(p_1)} [\![\boldsymbol{x}_2 \in \mathsf{supp}(p_2)]\!]$, $l(\boldsymbol{x}_2)|_{\mathsf{supp}(p_2)} [\![\boldsymbol{x}_1 \in \mathsf{supp}(p_1)]\!]$, and the additional support input unit $\log\theta_i[\![\boldsymbol{x} \in \mathsf{supp}(p_i)]\!]$ share the same support of $p$ by construction. Fig. 3 illustrates this property with one example. This implies that when a deterministic circuit and its logarithmic circuit are going to be multiplied, e.g., when computing entropies (Sec. C.2), we can check for their support to overlap in linear time (Alg. 3). $\quad\square$

*Proof of hardness.* We will prove hardness of computing the logarithm by showing hardness of computing the partition function of the logarithm of a circuit. Let $\mathbf{X} = \{X_1, \ldots, X_n\}$ be a collection of binary variables, and $p$ a smooth and decomposable PC over $\mathbf{X}$ where $p(\boldsymbol{x}) > 0$ for all $\boldsymbol{x} \in \mathsf{val}(\mathbf{X})$. Then computing the quantity

$$\sum_{\boldsymbol{x} \in \mathsf{val}(\mathbf{X})} \log p(\boldsymbol{x}) \tag{LOGPC}$$

is #P-hard.

The proof is by reduction from #NUMPAR, the counting problem of the number partitioning problem (NUMPAR) defined as follows. Given $n$ positive integers $k_1, \ldots, k_n$, we want to decide whether there exists a subset $S \subset [n]$ such that $\sum_{i \in S} k_i = \sum_{i \notin S} k_i$. NUMPAR is NP-complete, and #NUMPAR which asks for the number of solutions is known to be #P-hard.

We will show that we can solve #NUMPAR using an oracle for LOGPC, which will imply that LOGPC is also #P-hard. First, consider the following quantity SL for a given weight function $w(\cdot)$:

$$\mathsf{SL} := \sum_{\boldsymbol{x} \in \mathsf{val}(\mathbf{X})} \log(\sigma(w(\boldsymbol{x})) + 1) = \sum_{\boldsymbol{x} \in \mathsf{val}(\mathbf{X})} \log\left(\frac{1}{1 + e^{-w(\boldsymbol{x})}} + 1\right) = \sum_{\boldsymbol{x} \in \mathsf{val}(\mathbf{X})} \log\left(\frac{2 + e^{-w(\boldsymbol{x})}}{1 + e^{-w(\boldsymbol{x})}}\right)$$
$$= \sum_{\boldsymbol{x} \in \mathsf{val}(\mathbf{X})} \log(2 + e^{-w(\boldsymbol{x})}) - \sum_{\boldsymbol{x} \in \mathsf{val}(\mathbf{X})} \log(1 + e^{-w(\boldsymbol{x})}).$$

Similar to the construction in the proof of Thm. 3.4, we can construct smooth and decomposable, unnormalized PCs for $2 + e^{-w(\boldsymbol{x})}$ and $1 + e^{-w(\boldsymbol{x})}$ of size linear in $n$. Then, we can compute SL via two calls to the oracle for LOGPC on these PCs.

Next, we choose the weight function $w(\cdot)$ such that SL can be used to answer #NUMPAR. For a given instance of NUMPAR described by $k_1, \ldots, k_n$ and a large integer $m$, which will be chosen

later, we define the following weight function:

$$w(\boldsymbol{x}) := -\frac{m}{2} - m\sum_i k_i + 2m\sum_i k_i x_i.$$

In other words, $w(\boldsymbol{x}) = w_0 + \sum_i w_i x_i$ where $w_0 = -m/2 - m\sum_i k_i$ and $w_i = 2mk_i$ for $i = 1, \ldots, n$. Here, an assignment $\boldsymbol{x}$ corresponds to a subset $S_{\boldsymbol{x}} = \{i | x_i = 1, x_i \in \boldsymbol{x}\}$. Then the assignment $1 - \boldsymbol{x}$ corresponds to the complement $S_{1-\boldsymbol{x}} = \overline{S_{\boldsymbol{x}}}$. In the following, we will consider pairs of assignments $(\boldsymbol{x}, 1 - \boldsymbol{x})$ and say that it is a solution to NUMPAR if $S_{\boldsymbol{x}}$ and by extension $S_{1-\boldsymbol{x}}$ are solutions to NUMPAR.

Observe that if $(\boldsymbol{x}, 1 - \boldsymbol{x})$ is a solution to NUMPAR, then $w(\boldsymbol{x}) = w(1 - \boldsymbol{x}) = -m/2$. Otherwise, one of their weights must be $\geq m/2$ and the other $\leq -3m/2$. We can then deduce the following facts about the *contribution* of each pair to SL, defined as $c(\boldsymbol{x}, 1 - \boldsymbol{x}) = \log(\sigma(w(\boldsymbol{x})) + 1) + \log(\sigma(w(1 - \boldsymbol{x})) + 1)$.

If the pair $(\boldsymbol{x}, 1 - \boldsymbol{x})$ is a solution to NUMPAR, then its contribution to SL is going to be:

$$c(\boldsymbol{x}, 1 - \boldsymbol{x}) = 2\log(\sigma(-m/2) + 1).$$

Otherwise, we can bound its contribution as follows:

$$\log(\sigma(m/2) + 1) \leq c(\boldsymbol{x}, 1 - \boldsymbol{x}) \leq 1 + \log(\sigma(-3m/2) + 1)$$

If there are $k$ pairs that are solutions to the NUMPAR problem, then using the above observations we have the following bounds on SL:

$$\mathsf{SL} \geq (2^{n-1} - k)\log\left(\sigma(m/2) + 1\right) + 2k\log\left(\sigma(-m/2) + 1\right) \geq (2^{n-1} - k)\log\left(\sigma(m/2) + 1\right), \tag{1}$$

$$\mathsf{SL} \leq (2^{n-1} - k)(1 + \log\left(\sigma(-3m/2) + 1\right)) + 2k\log(\sigma(-m/2) + 1). \tag{2}$$

Suppose for some given $\epsilon > 0$, we select $m$ such that it satisfies both $1 - \epsilon \leq \log(\sigma(m/2) + 1)$ and $\log(\sigma(-m/2) + 1) \leq \epsilon$. First, this implies that $m$ also satisfies the following:

$$1 + \log\left(\sigma(-3m/2) + 1\right) \leq 1 + \log(\sigma(-m/2) + 1) \leq 1 + \epsilon.$$

Plugging in above inequalities to Eqs. (1) and (2), we get the following bounds on SL w.r.t. $\epsilon$ and $k$:

$$(2^{n-1} - k)(1 - \epsilon) \leq \mathsf{SL} \leq (2^{n-1} - k)(1 + \epsilon) + 2k\epsilon.$$

We can alternatively express this as the following bounds on $k$:

$$\frac{2^{n-1}(1 - \epsilon) - \mathsf{SL}}{1 - \epsilon} \leq k \leq \frac{2^{n-1}(1 + \epsilon) - \mathsf{SL}}{1 - \epsilon}.$$

The difference between the upper and lower bounds on $k$ is equal to $2^n \epsilon / (1 - \epsilon)$. If this difference is less than 1—e.g. by setting $\epsilon = 1/(2^n + 2)$—we can exactly solve for $k$. In particular, it must be equal to the ceiling of the lower bound as well as the floor of the upper bound. Moreover, the answer to #NUMPAR is given by $2k$. This concludes the proof that computing LOGPC is #P-hard. □

## B.6 Exponential Function of a Circuit

**Theorem 3.7** (Hardness of the exponential of a circuit). *Let $p$ be a smooth and decomposable circuit over variables $\mathbf{X}$. Then, computing its exponential $\exp(p(\mathbf{X}))$ as a decomposable circuit is #P-hard, even if $p$ is structured-decomposable.*

*Proof.* We will prove hardness of computing the exponential by showing hardness of computing the partition function of the exponential of a circuit. Let $\mathbf{X} = \{X_1, \ldots, X_n\}$ be a collection of binary variables with values in $\{-1, +1\}$ and let $p$ be a smooth and decomposable PC over $\mathbf{X}$ then computing the quantity

$$\sum_{\boldsymbol{x} \in \mathsf{val}(\mathbf{X})} \exp(p(\boldsymbol{x})) \tag{EXPOPC}$$

---

**Algorithm 6** LOGARITHM($p$, cache$_l$, cache$_s$)

---

1: **Input:** a smooth, deterministic and decomposable PC $p(\mathbf{X})$ and two caches for memoization (cache$_l$ for the logarithmic circuit and cache$_s$ for the support circuit).
2: **Output:** a smooth and decomposable circuit $l(\mathbf{X})$ encoding $\log(p(\mathbf{X}))$
3: **if** $p \in$ cache$_l$ **then return** cache$_l(p)$
4: **if** $p$ is an input unit **then**
5:     $l \leftarrow$ INPUT($\log\left(p_{|\mathsf{supp}(p)}\right), \phi(p)$)
6: **else if** $p$ is a sum unit **then**
7:     $n \leftarrow \{\}$
8:     **for** $i = 1$ **to** $|\mathsf{in}(p)|$ **do**
9:         $n \leftarrow n \cup \{\text{SUPPORT}(p_i, \mathsf{cache}_s)\} \cup \{\text{LOGARITHM}(p_i, \mathsf{cache}_l)\}$
10:     $l \leftarrow$ SUM($n, \{\log\theta_1, 1, \log\theta_2, 1, \ldots, \log\theta_{|\mathsf{in}(p)|}, 1\}$)
11: **else if** $p$ is a product unit **then**
12:     $n \leftarrow \{\}$
13:     **for** $i = 1$ **to** $|\mathsf{in}(p)|$ **do**
14:         $n \leftarrow n \cup \{\text{PRODUCT}(\{\text{LOGARITHM}(p_i, \mathsf{cache}_l)\} \cup \{\text{SUPPORT}(p_j, \mathsf{cache}_s)\}_{j \neq i})\}$
15:     $l \leftarrow$ SUM($n, \{1\}_{i=1}^{|\mathsf{in}(p)|}$)
16: cache$_l(p) \leftarrow l$
17: **return** $l$

---

is #P-hard.

The proof is a reduction from the problem of computing the partition function of an Ising model, ISING which is known to be #P-complete [20]. Given a graph $G = (V, E)$ with $n$ vertexes, computing the partition function of an Ising model associated to $G$ and equipped with potentials associated to its edges ($\{w_{u,v}\}_{(u,v) \in E}$) and vertexes ($\{w_v\}_{v \in V}$) equals to

$$\sum_{\boldsymbol{x} \in \mathsf{val}(\mathbf{X})} \exp\left(\sum_{(u,v) \in E} w_{u,v} x_u x_v + \sum_{v \in V} w_v x_v\right). \tag{ISING}$$

The reduction is made by constructing a smooth and decomposable circuit $p(\mathbf{X})$ that computes $\sum_{(u,v) \in E} w_{u,v} x_u x_v + \sum_{v \in V}$. This can be done by introducing a sum units with $|E| + |V|$ inputs that are product units and with weights $\{w_{u,v}\}_{(u,v) \in E} \cup \{w_v\}_{v \in V}$. The first $|E|$ product units receive inputs from $n$ input distributions where only 2 corresponds to the binary indicator inputs $X_u$ and $X_v$ for an edge $(u,v) \in E$ while the remaining $n-2$ are uniform distributions outputting 1 for all the possible states of variables $\mathbf{X} \setminus \{X_u, X_v\}$. Analogously, the remaining $|V|$ product units receive input from $n$ of which only one, corresponding to the vertex $v \in V$ is an indicator unit over $X_v$, while the remaining are uniform distributions for variables in $\mathbf{X} \setminus \{X_v\}$. $\qquad\square$

**Proposition 3.1** (Tractable exponential of a linear circuit). Let $p$ be a linear circuit over variables $\mathbf{X}$, i.e., $p(\mathbf{X}) = \sum_i \theta_i \cdot X_i$. Then $\exp(p(\mathbf{X}))$ can be represented as an omni-compatible circuit with a single product unit in $\mathcal{O}(|p|)$ time.

*Proof.* The proof follows immediately by the properties of exponentials of sums. Alg. 7 formalizes the construction. $\qquad\square$

---

**Algorithm 7** EXPONENTIAL($p$)

---

1: **Input:** a smooth circuit $p$ encoding $p(\mathbf{X}) = \theta_0 + \sum_{i=1}^n \theta_i X_i$
2: **Output:** its exponential circuit encoding $\exp(p(\mathbf{X}))$
3: $e \leftarrow \{\text{INPUT}(\exp(\theta_0 + \theta_1 X_1), X_1)\}$
4: **for** $i = 2$ **to** $n$ **do**
5:     $e \leftarrow e \cup \{\text{INPUT}(\exp(\theta_i X_i), X_i)\}$
6: **return** PRODUCT($e$)

---

## B.7 Other tractable operators over circuits

This section proves Lemma 3.8, which states that any operator over circuits that should yield a decomposable and smooth circuit as output must take the form of a sum, power, logarithm or exponential.

**Lemma 3.8** (Atlas Completeness). *Let $f$ be a continuous function. If (1) $f : \mathbb{R} \to \mathbb{R}$ satisfies $f(x + y) = f(x) + f(y)$ then it is a linear function $\beta \cdot x$; if (2) $f : \mathbb{R}_+ \to \mathbb{R}_+$ satisfies $f(x \cdot y) = f(x) \cdot f(y)$, then it takes the form $x^\beta$; if (3) instead $f : \mathbb{R}_+ \to \mathbb{R}$ satisfies $f(x \cdot y) = f(x) + f(y)$, then it takes the form $\beta \log(x)$; and if (4) $f : \mathbb{R} \to \mathbb{R}_+$ satisfies that $f(x + y) = f(x) \cdot f(y)$ then it is of the form $\exp(\beta \cdot x)$, for a certain $\beta \in \mathbb{R}$.*

*Proof.* The proof of all properties follows from constructing $f$ such that we obtain a *Cauchy functional equation* [21, 42].

The condition (1) exactly takes the form of a Cauchy functional equation, then it must hold that $f(x) = \beta \cdot x$.

For condition (2), let $g(x) = \log(f(\exp(x)))$ for all $x \in \mathbb{R}$, which is continuous because $f$ is. Then, it follows that

$$g(x + y) = \log(f(\exp(x + y))) = \log(f(\exp(x) \cdot \exp(y))) = \log(f(\exp(x))) + \log(f(\exp(y)))$$
$$= g(x) + g(y).$$

Therefore, $g(x)$ assumes the Cauchy functional form and, as in case (1), it is equal to $\beta \cdot x$. $\beta$ can be retrieved by solving $\beta \cdot x = \log(f(\exp(x)))$ for $x = 1$. This gives $\beta = \log(f(e))$. Applying the definition of $g$, we can hence write

$$f(\exp(x)) = e^{g(x)} = e^{\beta \cdot x} = (e^x)^\beta$$

Let $y \in \mathbb{R}_+$. Using the identity $y = e^{\log(y)}$ it follows that:

$$f(y) = f(e^{\log(y)}) = \left(e^{\log(y)}\right)^\beta = y^\beta.$$

Condition (3) follows an analogous pattern. Let $g(x) = f(\exp(x))$ for all $x \in \mathbb{R}$, which is continuous as $f$ is. Once again, $g$ satisfies the Cauchy functional form:

$$g(x + y) = f(\exp(x + y)) = f(\exp(x) \cdot \exp(y)) = f(\exp(x)) + f(\exp(y)) = g(x) + g(y).$$

Therefore, $g(x)$ must be of the form $\beta \cdot x$ for $\beta = f(e)$. Hence, $f(y) = \beta \log(y)$.

Lastly, for condition (4), $g(x) = \log(f(x))$ for all $x \in \mathbb{R}$, which is continuous if $f$ is. Then, we can retrieve the Cauchy functional by

$$g(x + y) = \log(f(x + y)) = \log(f(x) \cdot f(y)) = \log(f(x)) + \log(f(y)) = g(x) + g(y).$$

Therefore, $g(x)$ must be of the form $\beta \cdot x$. Hence, $f(y) = \exp(\beta \cdot y)$. $\qquad\square$

In summary, Lemma 3.8 states that if we want to enlarge our atlas beyond sum and product circuit operators, we need to focus our attention over powers, logarithms and exponentials. At the same time, it states that no operator with a different functional form and yet yielding a circuit made of sum and product units can be found. Extending our atlas to deal with a new language of circuits is an interesting future research direction.

## C Complex Information-Theoretic Queries

This section collects the complete tractability and hardness results for the queries in Tab. 2. Note that the tractability proofs are succinct thanks to our atlas which allows to define a tractable model class effortlessly. Some hardness proofs also benefit from the hardness results we provided for the simple operators in the previous section.

## C.1 Cross Entropy

**Theorem C.1.** *Let $p$ and $q$ be two compatible PCs over variables $\mathbf{X}$, and also let $q$ be deterministic. Then their cross-entropy, i.e.,*

$$-\int_{\mathsf{val}(\mathbf{X})} p(\boldsymbol{x}) \log(q(\boldsymbol{x})) d\mathbf{X},$$

*restricted to the support of $q$ can be exactly computed in $\mathcal{O}(|p|\,|q|)$ time. If $q$ is not deterministic, then computing their cross-entropy is #P-hard, even if $p$ and $q$ are compatible over $\mathbf{X}$.*

*Proof.* *(Tractability)* From Thm. 3.6 we know that we can compute the logarithm of $q$ in polytime, which is a PC of size $\mathcal{O}(|q|)$ that is compatible with $q$ and hence with $p$. Therefore, multiplying $p$ and $\log q$ according to Thm. 3.1 can be done exactly in polytime and yields a circuit of size $\mathcal{O}(|p|\,|q|)$ that is still smooth and decomposable, hence we can tractably compute its partition function.

*(Hardness)* The proof consists of a simple reduction from LOGPC from Thm. 3.6. We know that computing LOGPC for a smooth and decomposable PC over binary variables $\mathbf{X}$ is #P-hard. We can reduce this to computing the cross entropy between $p = 1$, which can be constructed as an omni-compatible circuit (Sec. A.2), and the original PC of the LOGPC problem. Thus, the cross-entropy of two compatible circuits is a #P-hard problem. $\square$

## C.2 Entropy

**Theorem C.2.** *Let $p$ be a smooth, deterministic, and decomposable PC over variables $\mathbf{X}$. Then its entropy,[9] defined as*

$$-\int_{\mathsf{val}(\mathbf{X})} p(\boldsymbol{x}) \log p(\boldsymbol{x})\, d\mathbf{X}$$

*can be exactly computed in $\mathcal{O}(|p|)$ time. If $p$ is smooth and decomposable but not deterministic, then computing its Shannon entropy, defined as*

$$\mathrm{ENT}(p) := -\sum_{\mathsf{val}(\mathbf{X})} p(\boldsymbol{x}) \log(p(\boldsymbol{x})) d\mathbf{X} \tag{ENTPC}$$

*is coNP-hard.*

*Proof.* *(Tractability)* Using Thm. 3.6 we can compute the logarithm of $p$ in polytime as a smooth and decomposable PC of size $\mathcal{O}(|p|)$ which furthermore shares the same support partitioning with $p$. Therefore, multiplying $p$ and $\log p$ according to Alg. 3 can be done in polytime and yields a smooth and decomposable circuit of size $\mathcal{O}(|p|)$ since $\log p$ shares the same support structure of $p$ (Thm. 3.6). Therefore, we can compute the partition function of the resulting circuit in time linear in its size.

*(Hardness)* The hardness proof contains a polytime reduction from the coNP-hard 3UNSAT problem, defined as follows: given a set of $n$ Boolean variables $\mathbf{X} = \{X_1, \ldots, X_n\}$ and a CNF with $m$ clauses $\{c_1, \ldots, c_m\}$ (each clause contains exactly 3 literals), decide whether the CNF is unsatisfiable.

The reduction borrows two gadget circuits $p_\beta$ and $p_\gamma$ defined in Sec. A.3. They each represent a logical formula over an auxiliary set of variables, which we denote here $\mathbf{X}'$, and thus outputs 0 or 1 for all values of $\mathbf{X}'$. Moreover, by construction, $p_\beta \cdot p_\gamma$ is the constant function 0 if and only if the original CNF is unsatisfiable.

We further construct a circuit $p_\alpha$ as the summation over $p_\beta$ and $p_\gamma$. Recall that $p_\beta$ and $p_\gamma$ can efficiently be constructed as smooth and decomposable circuits, and thus their sum can be represented as a smooth and decomposable circuit in polynomial time. We will now show that 3UNSAT can be reduced to checking whether the entropy of $p_\alpha$ is zero.

First, observe that for any assignment $\boldsymbol{x}'$ to $\mathbf{X}'$, $p_\alpha(\boldsymbol{x}')$ evaluates to 0, 1, or 2, because $p_\beta$ and $p_\gamma$ always evaluates to either 0 or 1. Moreover, if $p_\alpha$ only outputs 0 or 1 for all values of $\mathbf{X}'$, then $p_\beta \cdot p_\gamma$ must always be 0, implying that the original CNF is unsatisfiable. Lastly, in such a case, the entropy of $p_\alpha$ must be 0, whereas the entropy will be nonzero if there is an assignment $\boldsymbol{x}'$ such that

---

[9] For the continuous case this quantity refers to the *differential entropy*, while for the discrete case it is the Shannon entropy.

$p_\alpha(\boldsymbol{x}') = 2$. This concludes the proof that computing the entropy of a smooth and decomposable PC is coNP-hard. $\qquad\square$

## C.3  Mutual Information

**Theorem C.3.** *Let $p$ be a deterministic and structured-decomposable PC over variables $\mathbf{Z} = \mathbf{X} \cup \mathbf{Y}$ ($\mathbf{X} \cap \mathbf{Y} = \emptyset$). Then the mutual information between $\mathbf{X}$ and $\mathbf{Y}$, defined as*

$$\mathrm{MI}(p; \mathbf{X}, \mathbf{Y}) := \int_{\mathsf{val}(\mathbf{Z})} p(\boldsymbol{x}, \boldsymbol{y}) \log \frac{p(\boldsymbol{x}, \boldsymbol{y})}{p(\boldsymbol{x}) \cdot p(\boldsymbol{y})} d\mathbf{X} d\mathbf{Y},$$

*can be exactly computed in $\mathcal{O}(|p|)$ time if $p$ is still deterministic after marginalizing out $\mathbf{Y}$ as well as after marginalizing out $\mathbf{X}$.*[10] *If $p$ is instead smooth, decomposable, and deterministic, then computing the mutual information between $\mathbf{X}$ and $\mathbf{Y}$ is coNP-hard.*

*Proof. **(Tractability)** From Thm. 3.6 we know that the logarithm circuits of $p(\mathbf{X}, \mathbf{Y})$, $p(\mathbf{X})[\![\boldsymbol{y} \in \mathsf{supp}(p(\mathbf{Y}))]\!]$, and $p(\mathbf{Y})[\![\boldsymbol{x} \in \mathsf{supp}(p(\mathbf{X}))]\!]$ can be computed in polytime and are smooth and decomposable circuits of size $\mathcal{O}(|p|)$ that furthermore share the same support partitioning with $p(\mathbf{Y}, \mathbf{Z})$. Therefore, we can multiply $p(\mathbf{X}, \mathbf{Y})$ with each of these logarithm circuits efficiently according to Thm. 3.2 to yield circuits of size $\mathcal{O}(|p|)$. These are still smooth and decomposable circuits. Hence we can compute their partition functions and compute the mutual information between $\mathbf{X}$ and $\mathbf{Y}$ w.r.t. $p$.*

*(**Hardness**) We show hardness for the case of Boolean inputs, which implies hardness in the general case. This proof largely follows the hardness proof of Thm. C.2 to show that there is a polytime reduction from $\mathsf{3UNSAT}$ to the mutual information of PCs. For a given CNF, suppose we construct $p_\beta, p_\gamma$, and $p_\alpha = p_\beta + p_\gamma$ over a set of Boolean variables, say $\mathbf{X}$, as shown in Sec. A.2 and Thm. C.2.*

*Let $\mathbf{Y} = \{Y\}$ be a single Boolean variable, and define $p_\delta$ as:*

$$p_\delta := p_\beta \times [\![Y = 1]\!] + p_\gamma \times [\![Y = 0]\!].$$

*That is, we first construct two product units $q_1, q_2$ with inputs $\{p_\beta, [\![Y = 1]\!]\}$ and $\{p_\gamma, [\![Y = 0]\!]\}$, respectively, and build a sum unit $p_\delta$ with inputs $\{q_1, q_2\}$ and weights $\{1, 1\}$. Then $p_\delta$ has the following properties: **(1)** $p_\delta$ is smooth, decomposable, and deterministic, following from the fact that $p_\beta$ and $p_\gamma$ are also smooth, decomposable, and deterministic, and that $q_1$ and $q_2$ have no overlapping support. **(2)** $\mathrm{ENT}(p_\delta)$ can be computed in linear-time w.r.t. the circuit size by Thm. C.2. **(3)** $p_\delta(Y = 1)$ and $p_\delta(Y = 0)$ can be computed in linear time (w.r.t. size of the circuit $p_\delta$), as $p_\delta$ admits tractable marginalization. **(4)** For any $\boldsymbol{x} \in \mathsf{val}(\mathbf{X})$, $p_\delta(\boldsymbol{x}) = p_\beta(\boldsymbol{x}) + p_\gamma(\boldsymbol{x}) = p_\alpha(\boldsymbol{x})$.*

*We can express the mutual information $\mathrm{MI}(p_\delta; \mathbf{X}, \mathbf{Y})$ as:*

$$\mathrm{MI}(p_\delta; \mathbf{X}, \mathbf{Y}) = \mathrm{ENT}(p_\delta) - p_\delta(Y = 1) \log p_\delta(Y = 1) - p_\delta(Y = 0) \log p_\delta(Y = 0) - \mathrm{ENT}(p_\alpha).$$

*Therefore, given an oracle that computes $\mathrm{MI}(p_\delta; \mathbf{X}, \mathbf{Y})$, we can check if it is equal to $\mathrm{ENT}(p_\delta) - p_\delta(Y = 1) \log p_\delta(Y = 1) - p_\delta(Y = 0) \log p_\delta(Y = 0)$, which is equivalent to checking $\mathrm{ENT}(p_\alpha) = 0$, and decide whether the original CNF is unsatisfiable. Hence, computing the mutual information of smooth, deterministic, and decomposable PCs is a coNP-hard problem. $\qquad\square$*

## C.4  Kullback-Leibler Divergence

**Theorem C.4.** *Let $p$ and $q$ be two deterministic and compatible PCs over variables $\mathbf{X}$. Then, their intersectional Kullback-Leibler divergence (KLD), defined as*

$$\mathbb{D}_{\mathsf{KL}}(p \parallel q) = \int_{\mathsf{supp}(p) \cap \mathsf{supp}(q)} p(\boldsymbol{x}) \log \frac{p(\boldsymbol{x})}{q(\boldsymbol{x})} d\mathbf{X},$$

*can exactly be computed in $\mathcal{O}(|p| \, |q|)$ time. If $p$ and $q$ are not deterministic, then computing their KLD is #P-hard, even if they are compatible.*

---

[10]This structural property of circuits is also known as marginal determinism [8] and has been introduced in the context of marginal MAP inference and the computation of same-decision probabilities of Bayesian classifiers [35, 5].

*Proof. (**Tractability**)* Tractability of the intersectional KLD can be concluded directly from the tractability of cross entropy and entropy (Thm. C.1 and C.2). Specifically, KLD can be expressed as the difference between cross entropy and entropy:

$$\int p(\boldsymbol{x}) \log \frac{p(\boldsymbol{x})}{q(\boldsymbol{x})} \, d\mathbf{X} = \int p(\boldsymbol{x}) \log p(\boldsymbol{x}) \, d\mathbf{X} - \int p(\boldsymbol{x}) \log q(\boldsymbol{x}) \, d\mathbf{X}.$$

We can compute the entropy of a smooth, decomposable, and deterministic PC $p$ in $\mathcal{O}(|p|)$; and the cross entropy between two deterministic and compatible PCs $p$ and $q$ in $\mathcal{O}(|p|\,|q|)$ time.

*(**Hardness**)* The proof proceeds similarly to the hardness proof of Thm. C.1. Recall that the LOGPC problem from Thm. 3.6 is #P-hard for a smooth and decomposable PC over binary variables. We can reduce this to computing the negative of KL divergence between $p = 1$, which can be constructed as an omni-compatible circuit (Sec. A.2), and $q$ the original PC of the LOGPC problem. Thus, the KLD of two compatible circuits is a #P-hard problem. □

## C.5 Rényi Entropy

**Definition C.1** (Rényi entropy)**.** The Rényi entropy of order $\alpha \in \mathbb{R}$ of a PC $p$ is defined as

$$\frac{1}{1-\alpha} \log \int_{\mathsf{supp}(p)} p^{\alpha}(\boldsymbol{x}) d\mathbf{X}.$$

**Theorem C.5** (Rényi entropy for natural $\alpha$)**.** *Let $p$ be a structured-decomposable PC over variables $\mathbf{X}$ and $\alpha \in \mathbb{N}$. Its Rényi entropy can be computed in $\mathcal{O}(|p|^{\alpha})$ time. If $p$ is instead smooth and decomposable, then computing its Rényi entropy of order $\alpha$ is #P-hard.*

*Proof. (**Tractability**)* Tractability easily follows from computing the natural power circuit of $p$, which takes $\mathcal{O}(|p|^{\alpha})$ time according to Thm. 3.3.

*(**Hardness**)* We show hardness for the case of discrete inputs. The hardness of computing the Rényi entropy for natural number $\alpha$ is implied by the hardness of computing the natural power of smooth and decomposable PCs. Specifically, we conclude the proof by observing that there exists a polytime reduction from POW2PC, defined as $\sum_{\boldsymbol{x} \in \mathsf{val}(\mathbf{X})} p^2(\boldsymbol{x})$, a #P-hard problem as proved in Thm. 3.3, to Rényi entropy with $\alpha = 2$. □

**Theorem C.6** (Rényi entropy for real $\alpha$)**.** *Let $p$ be a smooth, decomposable, and deterministic PC over variables $\mathbf{X}$ and $\alpha \in \mathbb{R}_{+}$. Its Rényi entropy can be computed in $\mathcal{O}(|p|)$ time. If $p$ is not deterministic, then computing its Rényi entropy of order $\alpha$ is #P-hard, even if $p$ is structured-decomposable.*

*Proof. (**Tractability**)* Tractability easily follows from computing the power circuit of $p$, which takes $\mathcal{O}(|p|)$ time according to Thm. 3.5.

*(**Hardness**)* Similar to the hardness proof of Thm. C.5, this hardness result follows from the fact that computing the reciprocal of a structured-decomposable circuit is #P-hard (Thm. 3.4). Again, this is demonstrated by a polytime reduction from INVPC (i.e., $\sum_{\boldsymbol{x} \in \mathsf{val}(\mathbf{X})} p^{-1}(\boldsymbol{x})$) to Rényi entropy with $\alpha = -1$. □

## C.6 Rényi's $\alpha$-divergence

**Definition C.2** (Rényi's $\alpha$-divergence)**.** The Rényi's $\alpha$-divergence of two PCs $p$ and $q$ is defined as

$$\mathbb{D}_{\alpha}(p \parallel q) = \frac{1}{1-\alpha} \log \int_{\mathsf{supp}(p) \cap \mathsf{supp}(q)} p^{\alpha}(\boldsymbol{x}) q^{1-\alpha}(\boldsymbol{x}) d\mathbf{X}.$$

**Theorem C.7** (Hardness of alpha divergence of two PCs)**.** *Let $p$ and $q$ be two smooth and decomposable PCs over variables $\mathbf{X}$. Then computing their Rényi's $\alpha$-divergence for $\alpha \in \mathbb{R} \setminus \{1\}$ is #P-hard, even if $p$ and $q$ are compatible.*

*Proof.* Suppose $p$ is a smooth and decomposable PC $\mathbf{X}$ representing the constant function 1, which can be constructed as in Sec. A.2. Then $p^{\alpha}$ is also a constant 1. Hence, computing Rényi's 2-divergence between $p$ and another smooth and decomposable PC $q$ is as hard as computing the reciprocal of $q$, which is #P-hard (Thm. 3.4). □

**Theorem 4.1** (Tractable alpha divergence of two PCs). Let $p$ and $q$ be compatible PCs over variables $\mathbf{X}$. Then their Rényi's $\alpha$-divergence can be exactly computed in $\mathcal{O}(|p|^{\alpha}|q|)$ time for $\alpha \in \mathbb{N}, \alpha > 1$ if $q$ is deterministic or in $\mathcal{O}(|p||q|)$ for $\alpha \in \mathbb{R}, \alpha \neq 1$ if $p$ and $q$ are both deterministic.

*Proof.* The proof easily follows from first computing the power circuit of $p$ and $q$ according to Thm. 3.5 or Thm. 3.3 in polytime. Depending on the value of $\alpha$, the resulting circuits will have size $\mathcal{O}(|p|^{\alpha})$ and $\mathcal{O}(|q|)$ for $\alpha \in \mathbb{N}$ or $\mathcal{O}(|p|)$ and $\mathcal{O}(|q|)$ for $\alpha \in \mathbb{R}$ and will be compatible with the input circuits. Then, since they are compatible between themselves, their product can be done in polytime (Thm. 3.2) and it is going to be a smooth and decomposable PC of size $\mathcal{O}(|p|^{\alpha}|q|)$ (for $\alpha \in \mathbb{N}$) or $\mathcal{O}(|p||q|)$ (for $\alpha \in \mathbb{R}$), for which the partition function can be computed in time linear in its size. $\qquad\square$

## C.7 Itakura-Saito Divergence

**Theorem C.8.** *Let $p$ and $q$ be two deterministic and compatible PCs over variables $\mathbf{X}$, with bounded intersectional support $\mathsf{supp}(p) \cap \mathsf{supp}(q)$. Then their Itakura-Saito divergence, defined as*

$$\mathbb{D}_{\mathsf{IS}}(p \parallel q) = \int_{\mathsf{supp}(p)\cap\mathsf{supp}(q)} \left( \frac{p(\boldsymbol{x})}{q(\boldsymbol{x})} - \log\frac{p(\boldsymbol{x})}{q(\boldsymbol{x})} - 1 \right) d\mathbf{X}, \tag{3}$$

*can be exactly computed in $\mathcal{O}(|p||q|)$ time. If $p$ and $q$ are instead compatible but not deterministic, then computing their Itakura-Saito divergence is #P-hard.*

*Proof.* **(Tractability)** The proof easily follows from noting that the integral decomposes into three integrals over the inner sum: $\int_{\mathsf{supp}(p)\cap\mathsf{supp}(q)} \frac{p(\boldsymbol{x})}{q(\boldsymbol{x})} d\mathbf{X} - \int_{\mathsf{supp}(p)\cap\mathsf{supp}(q)} \log\frac{p(\boldsymbol{x})}{q(\boldsymbol{x})} d\mathbf{X} - \int_{\mathsf{supp}(p)\cap\mathsf{supp}(q)} 1\, d\mathbf{X}$.. Then, the first integral over the quotient can be solved $\mathcal{O}(|p||q|)$ (Thm. B.3); the second integral over the log of a quotient of two PCs can be computed in time $\mathcal{O}(|p||q|)$ (Thm. 3.6 and B.3) and finally the last one integrates to the dimensionality of $|\mathsf{supp}(p) \cap \mathsf{supp}(q)|$, which we assume to exist.

**(Hardness)** We show hardness for the case of binary variables $\mathbf{X} = \{X_1, \dots, X_n\}$. Suppose $q$ is an omni-compatible circuit representing the constant function 1, which can be constructed as in Sec. A.2. As such, integration in Eq. (3) becomes the summation $\sum_{\mathsf{val}(\mathbf{X})} p(\boldsymbol{x}) - \sum_{\mathsf{val}(\mathbf{X})} \log p(\boldsymbol{x}) - 2^n$. Hence, computing $\mathbb{D}_{\mathsf{IS}}$ must be as hard as computing $\sum_{\mathsf{val}(\mathbf{X})} \log p(\boldsymbol{x})$, since the first sum can be efficiently computed as $p$ must be smooth and decomposable by assumption and the last one is a constant. That is, we reduced the problem of computing the logarithm of the non-deterministic circuit (LOGPC, Thm. 3.6) to computing $\mathbb{D}_{\mathsf{IS}}$. $\qquad\square$

## C.8 Cauchy-Schwarz Divergence

**Theorem C.9.** *Let $p$ and $q$ be two structured-decomposable and compatible PCs over variables $\mathbf{X}$. Then their Cauchy-Schwarz divergence, defined as*

$$\mathbb{D}_{\mathsf{CS}}(p \parallel q) = -\log \frac{\int_{\boldsymbol{x}\in\mathsf{val}(\mathbf{X})} p(\boldsymbol{x})q(\boldsymbol{x})\, d\mathbf{X}}{\sqrt{\int_{\boldsymbol{x}\in\mathsf{val}(\mathbf{X})} p^2(x)\, d\mathbf{X} \int_{\boldsymbol{x}\in\mathsf{val}(\mathbf{X})} q^2(x)\, d\mathbf{X}}},$$

*can be exactly computed in time $\mathcal{O}(|p||q|+|p|^2+|q|^2)$. If $p$ and $q$ are instead structured-decomposable but not compatible, then computing their Cauchy-Schwarz divergence is #P-hard.*

*Proof.* **(Tractability)** The proof easily follows from noting that the numerator inside the log can be computed in $\mathcal{O}(|p||q|)$ time as a product of two compatible circuits (Thm. 3.2); and the integrals inside the square root at the denominator can both be solved in $\mathcal{O}(|p|^2)$ and $\mathcal{O}(|q|^2)$ respectively as natural powers of structured-decomposable circuits (Thm. 3.3).

**(Hardness)** The proof follows by noting that if $p$ and $q$ are structured-decomposable, then computing the denominator inside the log can be exactly done in $|p|^2 + |q|^2$ because they are natural powers of structured-decomposable circuits (Thm. 3.3). Then $\mathbb{D}_{\mathsf{CS}}$ must be as hard as a the product of two non-compatible circuits. Therefore we can reduce MULPC (Thm. 3.1) to computing $\mathbb{D}_{\mathsf{CS}}$. $\qquad\square$

## C.9   Squared Loss Divergence

**Theorem C.10.** *Let $p$ and $q$ be two structured-decomposable and compatible PCs over variables $\mathbf{X}$. Then their squared loss, defined as*

$$\mathbb{D}_{\mathsf{SL}}(p \parallel q) = \int_{\mathsf{val}(\mathbf{X})} \left( p(\boldsymbol{x}) - q(\boldsymbol{x}) \right)^2 d\mathbf{X},$$

*can be computed exactly in time $\mathcal{O}(|p|\,|q| + |p|^2 + |q|^2)$. If $p$ and $q$ are structured-decomposable but not compatible, then computing their squared loss is #P-hard.*

*Proof.* *(Tractability)* Proof follows by noting that the integral decomposes over the expanded square as $\int_{\mathsf{val}(\mathbf{X})} p^2(\boldsymbol{x})\, d\mathbf{X} + \int_{\mathsf{val}(\mathbf{X})} q^2(\boldsymbol{x})\, d\mathbf{X} - 2 \int_{\mathsf{val}(\mathbf{X})} p(\boldsymbol{x}) q(\boldsymbol{x})\, d\mathbf{X}$ and as such each integral can be computed by leveraging the tractable power of structured-decomposable circuits (Thm. 3.3) and the tractable product of compatible circuits (Thm. 3.2) and therefore the overall complexity is given by the maximum of the three.

*(Hardness)* Proof follows by noting that the integral decomposes over the expanded square as $\int_{\mathsf{val}(\mathbf{X})} p^2(\boldsymbol{x})\, d\mathbf{X} + \int_{\mathsf{val}(\mathbf{X})} q^2(\boldsymbol{x})\, d\mathbf{X} - 2 \int_{\mathsf{val}(\mathbf{X})} p(\boldsymbol{x}) q(\boldsymbol{x})\, d\mathbf{X}$ and that the first two terms can be computed in polytime as natural powers of structured-decomposable circuits (Thm. 3.3), hence computing $\mathbb{D}_{\mathsf{SL}}$ must be as hard as computing the product of two non-compatible circuits. Therefore we can reduce MULPC (Thm. 3.1) to computing $\mathbb{D}_{\mathsf{SL}}$. $\qquad\square$

# D   Expectation-based queries

This section completes the discussion around the complex queries that can be dealt with our atlas and details the expectations briefly discussed at the end of Sec. 4.

## D.1   Moments of a distribution

**Proposition D.1** (Tractable moments of a PC). *Let $p(\mathbf{X})$ be a smooth and decomposable PC over variables $\mathbf{X} = \{X_1, \ldots, X_d\}$, then for a set of natural numbers $\mathbf{k} = (k_1, \ldots, k_d)$, its $\mathbf{k}-$moment, defined as*

$$\int_{\mathsf{val}(\mathbf{X})} x_1^{k_1} x_2^{k_2} \ldots x_d^{k_d} p(\boldsymbol{x})\, d\mathbf{X}$$

*can be computed exactly in time $\mathcal{O}(|p|)$.*

*Proof.* The proof directly follows from representing $x_1^{k_1} x_2^{k_2} \ldots x_d^{k_d}$ as an omni-compatible circuit comprising a single product unit over $d$ input units, each encoding $x_i^{k_i}$, and then applying Cor. B.1. $\qquad\square$

## D.2   Probability of logical formulas

**Proposition D.2** (Tractable probability of a logical formula). *Let $p$ be a smooth and decomposable PC over variables $\mathbf{X}$ and $f$ an indicator function that represents a logical formula over $\mathbf{X}$ that can be compiled into a circuit compatible with $p$.[11] Then computing $\mathbb{P}_p[f]$ can be done in $\mathcal{O}(|p|\,|f|)$ time.*

*Proof.* It follows directly from Thm. 3.1, by noting that $\mathbb{P}_p[f] = \mathbb{E}_{\boldsymbol{x} \sim p(\mathbf{X})}[f(\boldsymbol{x})]$ and hence a tractable product between $p$ and $f$ suffices. $\qquad\square$

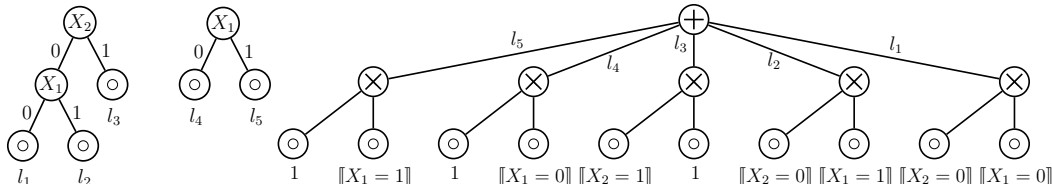

Figure 4: Encoding an additive ensemble of two trees over $\mathbf{X} = \{X_1, X_2\}$ (left) in an omni-compatible circuit over $\mathbf{X}$ (right).

### D.3 Expected predictions

**Example D.1** (Decision trees as circuits). *Let $\mathcal{F}$ be an additive ensemble of (decision or regression) trees over variables $\mathbf{X}$, also called a forest, and computing*

$$\mathcal{F}(\boldsymbol{x}) = \sum_{\mathcal{T}_i \in \mathcal{F}} \theta_i \mathcal{T}_i(\boldsymbol{x})$$

*for some input configuration $\boldsymbol{x} \in \mathsf{val}(\mathbf{X})$ and each $\mathcal{T}_i$ realizing a tree, i.e., a function of the form*

$$\mathcal{T}(\boldsymbol{x}) = \sum_{p_j \in \mathsf{paths}(\mathcal{T})} l_j \cdot \prod_{X_k \in \phi(p_j)} [\![x_k \leq \delta_k]\!]$$

*where the outer sum ranges over all possible paths in tree $\mathcal{T}$, $l_j \in \mathbb{R}$ is the label (class or predicted real) associated to the leaf of that path, and the product is over indicator functions encoding the decision to take one branch of the tree in path $p_j$ if $x_k$, the observed value for variable $X_k$ appearing in the decision node, i.e., satisfies the condition $[\![x_k \leq \delta_k]\!]$ for a certain threshold $\delta_k \in \mathbb{R}$.*

*Then, it is easy to transform $\mathcal{F}$ into an omni-compatible circuit $p(\mathbf{X})$ of the form*

$$p(\boldsymbol{x}) = \sum_{\mathcal{T}_i \in \mathcal{F}, p_j \in \mathsf{paths}(\mathcal{T}_)} l_j \cdot \prod_{X_k \in \phi(p_j)} [\![x_k \leq \delta_k]\!] \cdot \prod_{X'_k \notin \phi(p_j)} 1$$

*with a single sum unit realizing the outer sum and as many input product units as paths in the forest, each of which realizing a fully-factorized model over $\mathbf{X}$, and weighted by $l_j$. One example is shown in Fig. 4.*

**Proposition D.3** (Tractable expected predictions of additive ensembles of trees). *Let $p$ be a smooth and decomposable PC and $f$ an additive ensemble of $k$ decision trees over variables $\mathbf{X}$ and bounded depth. Then, its expected predictions can be exactly computed in $\mathcal{O}(k\,|p|)$.*

*Proof.* Recall that an additive ensemble of decision trees can be encoded as an omni-compatible circuit. Then, proof follows from Cor. B.1. $\qquad\square$

**Proposition D.4** (Tractable expected predictions of deep regressors (regression circuits)). *Let $p$ be a structured-decomposable PC over variables $\mathbf{X}$ and $f$ be a regression circuit [23] compatible with $p$ over $\mathbf{X}$, and defined as*

$$f_n(\boldsymbol{x}) = \begin{cases} 0 & \text{if } n \text{ is an input} \\ f_{n_\mathsf{L}}(\boldsymbol{x}_\mathsf{L}) + f_{n_\mathsf{R}}(\boldsymbol{x}_\mathsf{R}) & \text{if } n \text{ is an AND} \\ \sum_{c \in \mathsf{in}(n)} s_c(\boldsymbol{x})\,(\phi_c + f_c(\boldsymbol{x})) & \text{if } n \text{ is an OR} \end{cases}$$

*where $s_c(\boldsymbol{x}) = [\![\boldsymbol{x} \in \mathsf{supp}(c)]\!]$. Then, its expected predictions can be exactly computed in $\mathcal{O}(|p|\,|h|)$ time, where $h$ is its circuit representation as computed by Alg. 8.*

*Proof.* Proof follows from noting that Alg. 8 outputs a polysize circuit representation $h$ in polytime. Then, computing $\mathbb{E}_{\boldsymbol{x} \sim p(\mathbf{X})}[h(\boldsymbol{x})]$ can be done in $\mathcal{O}(|p|\,|h|)$ time by Thm. 3.2. $\qquad\square$

Table 4: Sizes of the intermediate and final circuits as processed by the operators in the pipelines of the Shannon and Rényi (for $\alpha = 1.5$) entropies and Kullback-Leibler and Alpha (for $\alpha = 1.5$) divergences when computed for two input circuits $p$ and $q$ learned from 20 different real-world datasets as in [11].

| DATASET | $p$ | $q$ | $p^\alpha$ | $q^{1-\alpha}$ | $r = \log(q)$ | $s = p/q$ | $t = \log(s)$ | $p \times q$ | $p \times r$ | $p \times t$ | $p^\alpha \times q^{1-\alpha}$ |
|---|---|---|---|---|---|---|---|---|---|---|---|
| NLTCS | 2779 | 7174 | 2779 | 7174 | 26155 | 7202 | 26239 | 7202 | 26183 | 26239 | 7202 |
| MSNBC | 2765 | 6614 | 2765 | 6614 | 24111 | 6634 | 24171 | 6634 | 24131 | 24171 | 6634 |
| KDD | 4963 | 50377 | 4963 | 50377 | 184575 | 50417 | 184695 | 50417 | 184615 | 184695 | 50417 |
| PLANTS | 12909 | 64018 | 12909 | 64018 | 234661 | 64070 | 234817 | 64070 | 234713 | 234817 | 64070 |
| AUDIO | 10278 | 45864 | 10278 | 45864 | 168062 | 45950 | 168320 | 45950 | 168148 | 168320 | 45950 |
| JESTER | 6475 | 35369 | 6475 | 35369 | 129579 | 35479 | 129909 | 35479 | 129689 | 129909 | 35479 |
| NETFLIX | 5068 | 14636 | 5068 | 14636 | 53571 | 14706 | 53781 | 14706 | 53641 | 53781 | 14706 |
| ACCIDENTS | 3193 | 8183 | 3193 | 8183 | 29891 | 8299 | 30239 | 8299 | 30007 | 30239 | 8299 |
| RETAIL | 4790 | 14926 | 4790 | 14926 | 54554 | 14994 | 54758 | 14994 | 54622 | 54758 | 14994 |
| PUMSB | 4277 | 12461 | 4277 | 12461 | 45500 | 12595 | 45902 | 12595 | 45634 | 45902 | 12595 |
| DNA | 73828 | 856955 | 73828 | 856955 | 3141981 | 857029 | 3142203 | 857029 | 3142055 | 3142203 | 857029 |
| KOSAREK | 5115 | 12988 | 5115 | 12988 | 47354 | 13106 | 47708 | 13106 | 47472 | 47708 | 13106 |
| MSNWEB | 4859 | 9025 | 4859 | 9025 | 32675 | 9175 | 33125 | 9175 | 32825 | 33125 | 9175 |
| BOOK | 7718 | 12731 | 7718 | 12731 | 45985 | 12943 | 46621 | 12943 | 46197 | 46621 | 12943 |
| MOVIE | 8309 | 11732 | 8309 | 11732 | 42374 | 11926 | 42956 | 11926 | 42568 | 42956 | 11926 |
| WEBKB | 10598 | 13397 | 10598 | 13397 | 47859 | 13653 | 48627 | 13653 | 48115 | 48627 | 13653 |
| CR52 | 10912 | 14348 | 10912 | 14348 | 51094 | 14546 | 51688 | 14546 | 51292 | 51688 | 14546 |
| C20NG | 11386 | 14630 | 11386 | 14630 | 52120 | 14886 | 52888 | 14886 | 52376 | 52888 | 14886 |
| BBC | 13884 | 17016 | 13884 | 17016 | 60857 | 17282 | 61655 | 17282 | 61123 | 61655 | 17282 |
| AD | 17744 | 21676 | 17744 | 21676 | 76870 | 21920 | 77602 | 21920 | 77114 | 77602 | 21920 |

Table 5: Times in seconds to compute the Shannon entropy (ENT), the cross-entropy (XENT), Kullback-Leibler (KLD), Alpha (for $\alpha = 1.5$) divergence, Rényi entropy (RényiEnt), and Cauchy-Schwarz divergence (CSDiv) over the circuits learned from 20 different real-world datasets by either using the algorithm distilled by our pipelines (see Tab. 4 and Fig. 5) compared to the custom and highly-optimized implementations of the same ENT [44] and KLD [28] algorithms as available in Juice.jl [12].

| DATASET | ENT | | KLD | | XENT | | ALPHADIV | | RÉNYIENT | | CSDIV | |
|---|---|---|---|---|---|---|---|---|---|---|---|---|
| | OURS | JUICE | OURS | JUICE | OURS | JUICE | OURS | JUICE | OURS | JUICE | OURS | JUICE |
| NLTCS | 0.143 | 0.001 | 0.830 | 0.207 | 0.422 | - | 0.140 | - | 0.013 | - | 0.300 | - |
| MSNBC | 0.109 | 0.001 | 0.369 | 0.182 | 0.297 | - | 0.105 | - | 0.018 | - | 0.227 | - |
| KDD | 0.157 | 0.001 | 3.154 | 0.790 | 2.180 | - | 0.885 | - | 0.016 | - | 1.136 | - |
| PLANTS | 0.679 | 0.005 | 3.983 | 3.909 | 3.739 | - | 1.160 | - | 0.088 | - | 1.572 | - |
| AUDIO | 0.406 | 0.003 | 2.736 | 1.681 | 1.873 | - | 0.537 | - | 0.029 | - | 0.771 | - |
| JESTER | 0.764 | 0.003 | 1.019 | 0.432 | 0.805 | - | 0.351 | - | 0.024 | - | 0.476 | - |
| NETFLIX | 0.106 | 0.002 | 0.352 | 0.175 | 0.264 | - | 0.100 | - | 0.017 | - | 0.201 | - |
| ACCIDENTS | 0.055 | 0.001 | 0.207 | 0.039 | 0.542 | - | 0.091 | - | 0.009 | - | 0.124 | - |
| RETAIL | 0.108 | 0.001 | 0.508 | 0.153 | 0.415 | - | 0.184 | - | 0.013 | - | 0.197 | - |
| PUMSB | 0.092 | 0.001 | 0.701 | 0.133 | 0.316 | - | 0.119 | - | 0.012 | - | 0.214 | - |
| DNA | 4.365 | 0.027 | 64.664 | 220.377 | 52.997 | - | 15.609 | - | 0.255 | - | 22.901 | - |
| KOSAREK | 0.182 | 0.002 | 0.477 | 0.106 | 0.379 | - | 0.139 | - | 0.011 | - | 0.735 | - |
| MSNWEB | 0.128 | 0.002 | 0.261 | 0.047 | 0.211 | - | 0.342 | - | 0.015 | - | 0.135 | - |
| BOOK | 0.086 | 0.003 | 0.215 | 0.036 | 0.202 | - | 0.075 | - | 0.020 | - | 0.115 | - |
| MOVIE | 0.272 | 0.002 | 0.443 | 0.063 | 0.373 | - | 0.172 | - | 0.015 | - | 0.194 | - |
| WEBKB | 0.138 | 0.003 | 0.241 | 0.031 | 0.164 | - | 0.079 | - | 0.023 | - | 0.098 | - |
| CR52 | 0.141 | 0.004 | 0.260 | 0.035 | 0.188 | - | 0.087 | - | 0.031 | - | 0.143 | - |
| C20NG | 0.118 | 0.003 | 0.264 | 0.034 | 0.194 | - | 0.088 | - | 0.032 | - | 0.101 | - |
| BBC | 0.205 | 0.005 | 0.308 | 0.037 | 0.225 | - | 0.110 | - | 0.038 | - | 0.189 | - |
| AD | 0.193 | 0.007 | 0.346 | 0.046 | 0.281 | - | 0.151 | - | 0.031 | - | 0.207 | - |

---

**Algorithm 8** $\text{RGCTOCIRCUIT}(r, \text{cache}_r, \text{cache}_s)$

---

1: **Input:** a regression circuit $r$ over variables $\mathbf{X}$ and two caches for memoization (i.e., $\text{cache}_r$ and $\text{cache}_s$).
2: **Output:** its representation as a circuit $p(\mathbf{X})$.
3: **if** $r \in \text{cache}_r$ **then return** $\text{cache}_r(r)$
4: **if** $r$ is an input gate **then**
5: $\quad p \leftarrow \text{INPUT}(0, \phi(r))$
6: **else if** $r$ is a sum gate **then**
7: $\quad n \leftarrow \{\}$
8: $\quad$ **for** $i = 1$ **to** $|\text{in}(r)|$ **do**
9: $\qquad n \leftarrow n \cup \{\text{SUPPORT}(r_i, \text{cache}_s)\} \cup \{\text{RGCTOCIRCUIT}(r_i, \text{cache}_r)\}$
10: $\quad p \leftarrow \text{SUM}(n, \{\theta_i, 1_1, \ldots, 1_{\text{in}(p)}\}_{i=1}^{|\text{in}(r)|})$
11: **else if** $r$ is a product gate **then**
12: $\quad$ **for** $i = 1$ **to** $|\text{in}(r)|$ **do**
13: $\qquad p \leftarrow \text{PRODUCT}(\{\text{RGCTOCIRCUIT}(r_i, \text{cache}_r)\} \cup \{\text{SUPPORT}(r_j, \text{cache}_s)\}_{j \neq i})$
14: $\text{cache}_r(r) \leftarrow p$
15: **return** $p$

---

```
function kld(p, q)                function xent(p, q)              function csdiv(p, q)
    r = quotient(p, q)               r = log(q)                       r = product(p, q)
    s = log(r)                       s = product(p, r)                s = real_pow(p, 2.0)
    t = product(p, s)                return -integrate(s)             t = real_pow(q, 2.0)
    return integrate(t)          end                                  a = integrate(r)
end                                                                   b = integrate(s)
                                                                      c = integrate(t)
                                                                      return -log(a / sqrt(b * c))
function ent(p)                   function alphadiv(p, q, alpha=1.5)  end
    q = log(p)                        r = real_pow(p, alpha)
    r = product(p, q)                 s = real_pow(q, 1.0-alpha)
    return -integrate(s)              t = product(r, s)
end                                   return log(integrate(t)) / (1.0-alpha)
                                  end
```

Figure 5: The modular operators defined in Sec. 3 can be easily composed to implement tractable algorithms for novel query classes. Here we show the code snippet for five queries: Kullback-Leibler divergence (`kld`), Cross Entropy (`xent`), Entropy (`ent`), Alpha divergence (`alphadiv`), and Cauchy-Schwarz divergence (`csdiv`).

# E  Experiments

**Generated PCs** All adopted PCs were generated by running Strudel [11] on the twenty density estimation benchmarks [48]. For every dataset, we ran Strudel twice with 200 and 500 iterations, respectively. All other hyperparameters were selected following Dang et al. [11].

**Server specifications** All our experiments were run on a server with 72 CPUs, 512G Memory, and 2 TITAN RTX GPUs.

**Implementations** Code snippet for the five adopted queries (i.e., Kullback-Leibler divergence, Cross Entropy, Entropy, Alpha divergence, and Cauchy-Schwarz divergence) are shown in Fig. 5. Note that they are simple compositions of the modular operators introduced in Sec. 3.

---

[11]E.g. by compiling it into an SDD [14, 3] whose vtree encodes the hierarchical scope partitioning of $p$.