# OpenReview forum: "A Compositional Atlas of Tractable Circuit Operations for Probabilistic Inference"
_NeurIPS.cc/2021/Conference — NeurIPS 2021 Oral_

### Official Review · Reviewer_cgs4 · 2021-07-12

**Rating:** 8
**Confidence:** 2

**Summary:**

This paper investigate the tractability of arithmetic operations on circuits for probabilistic inference.  There is considerable similar work when it comes to operators on logical circuits, and this paper generalises this to operators such as powers, logarithm and exponentials. The paper also considers these operations in a computational graph, which means that the results of this paper can be used to prove tractability conditions for many queries, in particular divergences and entropy calculations. The paper discusses various examples of such tractability results.


**Limitations And Societal Impact:**

Societal impact is not relevant for this paper. The authors do address various hardness results, which are limitations for probabilistic inference in general. This is fine.

**Main Review:**

The paper is well-written, and despite its technical nature, is quite easy to follow. The main ideas are discussed in the paper, and sometimes intuitive ideas behind proofs are explained. I thought the general ideas were clear.

The key concept that seems to be introduced in this paper is compatibility, which turns out to be important to prove many of the tractability results. As far as I can see, this idea is novel and certainly non-trivial. In my estimation, this is a major step forward in this research area. I would expect that this will have significant impact in designing new methods that rely on tractable computations of information-theoretic queries.

As far as I can see, the paper is technically sound. The experimental results are obviously limited for this type of paper. Some preliminary observations are made from experiments, which I think is sufficient at this point.

**Time Spent Reviewing:**

3

---

> ### Author Response · Authors · 2021-08-09
> **Thanks for seeing our work as a major step forward this research area**
>
> We thank the reviewer for the time they spent reading, reviewing our work and for evaluating it to be technically sound, novel, and a major step forward in tractable inference.

---

### Official Review · Reviewer_gLaG · 2021-07-15

**Rating:** 7
**Confidence:** 2

**Summary:**

This paper introduces a general framework to trace the tractability of complex queries involving circuit representations in a unified manner over model classes and query classes, and to automatically distill tractable inference functions. Using the framework, the paper unifies and generalizes existing inference algorithms in the literature, and presents new tractability and hardness results of many information-theoretic queries.

**Limitations And Societal Impact:**

Yes

**Main Review:**

The proposed framework is novel, and can be very useful for understanding the tractability of complex queries involving circuit representations, as well as deriving tractable inference algorithms. I did not check all the details in the proofs, but the paper seems technically sound, and is clearly written and well organized. The empirical results showing the practicality of the framework is a nice addition.

Some questions to the authors/that need more clarifications:
1. Is there any redundancy involved in the automatically derived inference algorithms? In other words, are there still advantages in some of the manually derived inference algorithms, even through they are essentially calculating the same thing? What are some sources of this redundancy? There are some brief discussions in L370-377 but it would be informative to include a brief discussion on how the distilled inference algorithm and the highly optimized implementation differ.
2. It looks like most of the results give sufficient conditions. To what extent are these conditions also necessary? In other words, to what extent is the framework complete? And what are some cases that are not covered by the framework?

=================================

Update after rebuttal: I thank the authors for the clarifications. I still think this is a good paper and would recommend acceptance.

**Time Spent Reviewing:**

3

---

> ### Author Response · Authors · 2021-08-09
> **Thank you for recognizing the usefulness of our framework**
>
> We thank the reviewer for the time invested in reviewing and for judging our work novel, potentially useful and technically sound.
>
> > are there still advantages in some of the manually derived inference algorithms, even through they are essentially calculating the same thing
>
> The major difference between implementations is that in our prototypes intermediate circuits are always materialized in memory. On the other hand, an optimized implementation, such as that for the entropy query, can avoid this by operating in place on the same input circuits. In fact, avoiding costly memory allocations is the main factor responsible for reducing execution times by one-two orders of magnitude, see Table 4 in the Appendix. Additional optimizations in complex query implementations might involve using a global cache to store partial computations across all operators (e.g., to compute support circuits), while our current implementation does not. We point out that both optimizations could be introduced in our framework. This effort would be akin to crafting a compiler for fast inference: given a query pipeline we can perform a light static analysis to determine which data structures (circuits and caches) are worth being shared across operators.
>
> > To what extent are these conditions also necessary?
>
> In a broad sense, certain structural properties are “necessary” to guarantee tractability of a query within our framework: when these properties are unmet, some operators become suddenly hard when they have to compute smooth and decomposable circuits as outputs, as shown in Tables 1 and 2. For example, the logarithm operator can output a smooth and decomposable circuit in polytime when its input is smooth, decomposable and deterministic; when determinism is missing from the input, the operation becomes #P-hard.
>
> For a characterization of necessary conditions that do not depend on P!=NP, for certain query pipelines such as the computation of moments of circuits, we could adopt a proving strategy analogous to that used by Choi et al. (2020) to demonstrate Proposition 2.1 (as referenced in our paper).

---

### Official Review · Reviewer_fFgc · 2021-07-16

**Rating:** 7
**Confidence:** 3

**Summary:**

A general framework for developing tractable models using circuits. The main contributions are defining the building blocks for tractable models such that queries can be expressed as compositions of these blocks and also open-source software for building these models quickly with tractable guarantees.

**Limitations And Societal Impact:**

It is hard to see any potential negative societal impact with this work since it seems quite foundational in nature.

**Main Review:**

The paper proposes a general framework for tractable queries based on a circuit representation. The main contribution is to abstract out a set of operations and new queries that can be composed using these abstractions are provably tractable. Thus, new tractable algorithms can be easily developed by composing the operations. Tractability conditions and hardness results are proved for operations. The results are applied to prove hardness or tractability results for several common queries. The main takeaway is that these proofs are quite easy since they build upon the basic operations. It is shown that some information theoretic queries for which tractable algorithms were not available before can be tractably computed using the proposed framework. Experiments are performed as part of an open source package to demonstrate the practical implementation of tractable algorithms.

The paper is very well written and clear in terms of contributions. While it does cover a lot of material given the page limit, the organization seems very nice with details in the appendix. In terms of overall significance, having the abstraction for tractable models is a valuable contribution along with an open source software package that one can use to quickly implement tractable models. This could potentially have impact in various important sub-fields such as XAI and Fairness in AI.

Overall, I did not see many weaknesses in this paper. One aspect that could be improved is if algorithms may not fit into the proposed tractable framework through the defined operations, some comments on how they could be approximated in the pipeline may also be useful for practical relevance.

**Time Spent Reviewing:**

4

---

> ### Author Response · Authors · 2021-08-09
> **Thanks for recognizing the impact our paper can have**
>
> We thank the reviewer for deeming our paper well organized and potentially impacting important sub-fields in AI.
>
>
> > some comments on how they could be approximated in the pipeline may also be useful for practical relevance
>
> We agree that dealing with approximations is an interesting future research venue, which becomes possible only now that our framework laid the foundations.
>
> One way to tackle this problem could be to derive ways to relax or enforce the structural properties of an input circuit to obtain a new circuit that would encode an upper and/or lower bound of the original one. This interesting direction is definitely challenging: it might not be possible to obtain approximations with guarantees for certain queries. We note, in fact, that the inapproximability of some queries might be derived by the hardness of approximating certain operators. Some of the hardness reductions we provided suggest this scenario.
>
> Nevertheless, we plan to investigate coarser, practical approximations (heuristics) through optimization schemes, a-la-variational inference.  Our framework could serve as a basis for compressing or distilling circuits with certain structural properties out of other circuits classes.

---

### Official Review · Reviewer_FfYX · 2021-07-17

**Rating:** 7
**Confidence:** 3

**Summary:**

This work suggests a general framework for deriving complex information-theoretic queries using probabilistic circuits (PCs). This framework represents a complex query as a pipeline of operations. The manuscript also provides algorithms for applying these operations to PCs as well.

**Limitations And Societal Impact:**

The paper does not explicitly discuss the limitations of the general framework. It would be interesting to see possible assumptions required for adopting this unifying perspective using PCs.

**Main Review:**

This work is original in its unifying characteristic. The contributions presented here provide a simple and helpful tool for tractability analysis of common queries in Machine Learning (ML) and related fields.

The manuscript has an excellent technical quality. The theoretical contributions are non-trivial. Moreover, the work discusses different cases, such as deterministic and non-deterministic PCs. Derived theoretical results are sound, to the best of my knowledge.

This paper is well-written, and it has a clear presentation. For instance, Table 1 summarizes sufficient conditions for the tractability of typical circuit operations.

This work is relevant to a broader audience besides the PC community. Indeed, the manuscript summarizes in Table 2 tractability results for various commonly used machine learning functions.

**Time Spent Reviewing:**

1

---

> ### Author Response · Authors · 2021-08-09
> **Thanks for the positive feedback**
>
> Thanks for the time spent reviewing our work and for praising its technical quality and presentation.
>
>
>   > The paper does not explicitly discuss the limitations of the general framework
>
> We agree that showing limitations of a framework is important. This is why we introduced several hardness results in our work (Table 1 and 2): they are a type of analysis of how far we can push the boundaries of tractability for circuit operators and queries.

---

### Decision · Program_Chairs · 2021-09-27

**Decision:**

Accept (Oral)

**Comment:**

Very interesting solid work with potential impact. Please take all comments in consideration.